# Information-Theoretic Questionnaire Construction for Consistent Evaluation of Subjective Tasks with LLMs

## Abstract

Despite the growing use of large language models (LLMs) in subjective tasks such as role-playing, humor, emotional intelligence, and dialogue quality, their evaluation faces a pressing **reproducibility crisis**: even the same evaluator may contradict itself when re-judging the exact same sample. We attribute this instability to dimension drift, where free-form evaluation protocols (e.g., Chain-of-Thought reasoning) unpredictably shift their implicit criteria, undermining reliability. To address this fundamental challenge, we reformulate subjective evaluation as an information-theoretic optimization problem. Specifically, we propose an **Expected Information Gain (EIG)-based framework** that constructs a stable yet adaptive personalized rubric to eliminate dimension drift. Our two-stage "generate–then–score" design first produces a diverse pool of candidate evaluation questions and then selects the most informative subset via EIG, yielding explicit and repeatable criteria. Experiments on six benchmarks, including CharacterEval, The rJokes, and MT_bench, demonstrate that our approach substantially improves both evaluation consistency and alignment with human judgments, outperforming CoT-based and fixed-questionnaire baselines. These results highlight that information-theoretic questionnaire construction offers a principled and reliable path toward reproducible evaluation of subjective tasks.

## 1 Introduction

Large language models (LLMs) are increasingly applied to tasks with inherently subjective dimensions, including role-playing Tu et al. (2024), humor understanding Narad et al. (2025), emotional intelligence Paech (2023), and dialogue evaluation Ou et al. (2024); Bai et al. (2024). With their growing capabilities, LLMs are no longer used solely as task performers, but also as evaluators in so-called "LLM-as-a-judge" frameworks, where they are expected to provide scalable, low-cost alternatives to human annotation. However, alongside this trend emerges an even more troubling phenomenon: *intra-annotator inconsistency*. The same evaluator may assign different scores, or even shift its evaluation criteria, when asked to re-judge the exact same sample. Such self-contradiction under identical conditions exposes a *reproducibility crisis*: even the same judge, faced with the same input, can contradict itself in both scores and rationales. This instability not only undermines reliability and fairness, but also cripples the ability to track genuine progress, making it a central bottleneck that threatens the comparability of research outcomes across the community.

Empirical evidence highlights how pervasive this crisis is. We identify the underlying mechanism as *dimension drift*: across repeated trials, the evaluation criteria articulated by the same judge shift unpredictably, causing judgments to vary even on identical samples. This drift stems from a fundamental weakness of free-form protocols: without a fixed rubric, evaluators can arbitrarily introduce or discard dimensions, causing unstable and inconsistent judgments. Among these protocols, chain-of-thought (CoT) reasoning in the style of G-Eval Liu et al. (2023) is the most prominent, and strikingly, also one of the most vulnerable. As shown in Fig. 1(a), CoT exhibits lower repeatability than even a simple "Direct" method, while Fig. 1(b) provides a concrete illustration: the same judge focuses on only partially overlapping dimensions across two identical evaluations, leading to inconsistent outcomes. This combination of quantitative and qualitative evidence underscores that dimension drift is not anecdotal, but systemic.

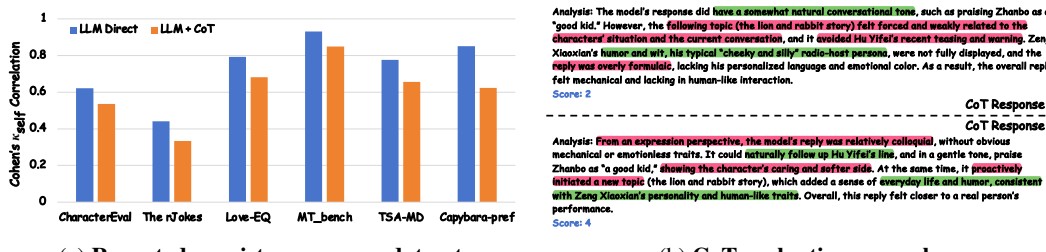

(a) **Repeated-consistency on our dataset.**    (b) **CoT evaluation example.**

Figure 1: Repeatability analysis for LLM-as-a-judge. Panel (a) compares Direct vs. CoT under repeated evaluations on the same samples; Panel (b) shows overlap (green) and omissions/additions (red) of articulated criteria across CoT repeats.

Existing approaches have tried to address this instability but remain unsatisfactory. Multi-judge and ensemble strategies such as LMC Zhao et al. (2024), PoLL Verga et al. (2024), or CompassJudger Cao et al. (2024) enhance robustness through aggregation or specialized training, but incur high computational cost and still fail to prevent drift at the individual judge level. Fixed-questionnaire methods such as CheckEval Pereira et al. (2024) enforce consistent rubrics and thus eliminate drift, but their static design makes them brittle and insensitive to task- or sample-specific nuances. In short, the field faces a dilemma: free-form protocols (e.g., CoT) are flexible but unstable, while fixed rubrics are stable but rigid.

How, then, can we stabilize evaluation without sacrificing task relevance? We argue for a *stability–informativeness loop*: (1) a stable rubric to eliminate dimension drift and ensure reproducibility, and (2) a principled mechanism to select the most *informative* questions so that evaluations remain adaptive and task-specific. In this framing, inconsistency arises when evaluation is *rubric-blind*, while brittleness arises when it is *rubric-rigid*. The ideal solution must therefore achieve both **stability** and **adaptivity** in a single design.

To this end, we propose a novel **Expected Information Gain (EIG) questionnaire framework**. We reformulate subjective evaluation as an information-theoretic optimization problem. Information theory provides an elegant principle for identifying "good" evaluation questions, defined as those whose answers maximally reduce uncertainty about the final judgment. In this view, stability and adaptivity are not conflicting goals, but two sides of the same principle: by quantifying informativeness, we can construct rubrics that are simultaneously reproducible and task-sensitive. Concretely, our loop operates as follows: (i) generate a diverse pool of candidate evaluation questions, ensuring the potential for adaptivity; (ii) estimate a simulated joint distribution of answers and scores on unlabeled samples, and select the subset of questions maximizing EIG, producing a rubric that is compact yet maximally informative; (iii) use this rubric to guide subsequent evaluations, thereby enforcing consistency across trials while retaining sensitivity to task-specific nuances. This stability–informativeness loop avoids the arbitrariness of CoT reasoning while overcoming the rigidity of fixed questionnaires, offering a principled and general solution to reproducible subjective evaluation.

Our contributions are threefold:

- We **identify and systematically define** the problem of *dimension drift*, establishing intra-annotator inconsistency as a central bottleneck for subjective evaluation in LLM-as-a-judge settings, and introduce an explicit, task-specific questionnaire paradigm to mitigate it.

- We **reformulate subjective evaluation as an information-theoretic optimization problem**, introducing a paradigm shift that reconciles stability and adaptivity by selecting questions that maximize Expected Information Gain (EIG).

- We instantiate this principle in a practical stability–informativeness loop, and validate it on six benchmarks, achieving substantial gains over CoT-based and fixed-rubric baselines. In particular, our approach improves evaluation self-consistency by up to **7.6% in Cohen's** $\kappa$, while also enhancing alignment with human judgments.

## 2 RELATED WORK

**Evaluation Paradigms and the Stability Challenge.** Evaluating subjective tasks such as role-playing, humor, or creative writing is inherently difficult. Traditional metrics like BLEU Papineni et al. (2002), ROUGE Lin (2004), METEOR Banerjee & Lavie (2005), CIDEr Vedantam et al. (2015), and embedding-based scores such as BERTScore Zhang et al. (2019) or MoverScore Zhao et al. (2019) capture surface overlap but miss deeper stylistic and pragmatic qualities. Human evaluation remains the "gold standard," yet is costly, slow, and inconsistent Callison-Burch et al. (2007); Novikova et al. (2017). These challenges are especially acute in creative and open-ended tasks Deriu et al. (2020); Wieting et al. (2022). Against this backdrop, LLM-as-a-Judge has emerged as a scalable alternative. Models such as GPT-4, prompted with rubrics as in G-Eval Liu et al. (2023), *sacrifice stability for adaptivity*: while flexible, they suffer from dimension drift and low repeatability. Checklist methods such as CheckEval Lee et al. (2024) enforce *stability* but at the cost of *adaptivity* to task-dependent or instance-specific subtleties. Multi-judge ensembles such as LMC Zhao et al. (2024), PoLL Verga et al. (2024), or CompassJudger Cao et al. (2024) seek robustness through aggregation, but only by increasing computational cost; each constituent judge still shifts criteria, leaving the core problem unsolved. In short, existing paradigms split between flexible but unstable protocols and stable but rigid checklists. This unresolved trade-off between stability and adaptivity motivates the need for a principled framework.

**Rubric Design for Subjective Evaluation** A complementary line of work examines how rubrics themselves are constructed. Fixed rubrics ensure consistency but often miss task-specific nuances. For example, Wei et al. Wei et al. (2024) show that varying prompt templates or evaluation dimensions can alter both human-model agreement and retest stability, while Liu et al. Liu et al. (2025) highlight the need to explicitly model judge consistency. These findings underscore that rubric design is not neutral: the chosen dimensions fundamentally shape evaluation outcomes. Current methods sit at two extremes: fixed rubrics provide stability but are brittle, while free-form reasoning allows adaptivity but suffers from dimension drift. Neither offers a principled way to select evaluation dimensions. In contrast, information-theoretic principles such as Expected Information Gain (EIG), widely applied in active learning Felder & Brent (2009) and Bayesian optimization Frazier (2018), provide a natural solution. Our work introduces EIG into rubric construction for subjective LLM evaluation, transforming it from a heuristic process into a principled optimization problem. This enables rubrics that are both stable across trials and adaptive to task-specific nuances, resolving the stability–adaptivity trade-off left open by prior work.

## 3 METHODOLOGY

We improve the reliability of subjective evaluation by explicitly materializing rubrics as questionnaires, instead of relying on implicit and unstable criteria, as shown in Figure 2. Grounded in Bayesian Experimental Design (BED), each evaluation question is treated as an experiment whose utility is measured by its expected reduction of uncertainty about the final rating $y$. This reformulates subjective evaluation as an *information-theoretic optimization problem*, where the goal is to select a compact set of questions that maximally reduce entropy. Our methodology proceeds in five stages: **(1) Framework and Formalism (Section 3.1)**: define evaluation instances, rating space, and the target distribution $p(y \mid I)$. **(2) Candidate Generation (Section 3.2)**: build a diverse, high-entropy pool $Q$ of evaluation questions. **(3) Joint Distribution Estimation (Section 3.3)**: approximate $p(y, a \mid q, I)$ through LLM simulations. **(4) Expected Information Gain (Section 3.4)**: compute EIG for each question and select the most informative subset $Q^*$. **(5) Final Evaluation (Section 3.5)**: prompt the LLM-as-a-judge with $Q^*$ and aggregate answers into stable ratings. The full algorithm is provided in Appendix A.3.

This stability–informativeness loop ensures that $Q^*$ is both adaptive and reproducible: candidate diversity provides flexibility, EIG quantifies informativeness, and the fixed rubric anchors consistent evaluation, eliminating the dimension drift of free-form protocols.

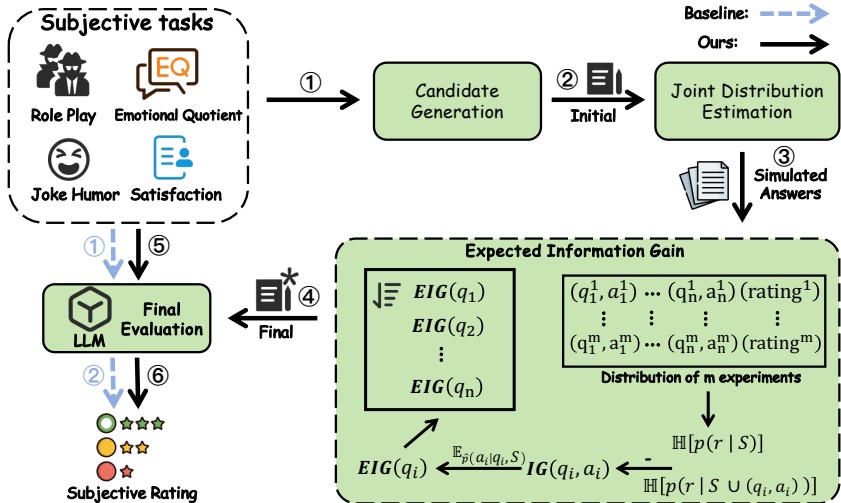

Figure 2: Overview of our framework. We construct task-specific questionnaires by generating candidate evaluation questions, estimating their informational value via simulated responses, selecting the most informative subset using Expected Information Gain (EIG), and finally conducting questionnaire-style LLM-as-a-judge evaluation.

## 3.1 Framework and Formalism

At a high level, our framework views evaluation as an information-gathering process: there exists an "ideal" score for each model response, but it is hidden. To approximate it, we ask a few structured questions whose answers gradually reduce uncertainty about the score. The key problem then becomes:which questions are most informative. This intuition motivates the formal definitions below.

**Evaluation Instance:** Each evaluation instance is a tuple $I = (S, M)$, where $S$ denotes the evaluation context and $M$ is the model's response under evaluation. The framework is agnostic to task specifics: $S$ may encode a role profile, a dialogue history, or any other contextual information depending on the evaluation domain.

**Rating:** A rating $y \in \mathcal{Y}$ (e.g., $\mathcal{Y} = \{1, 2, 3, 4, 5\}$) represents the overall subjective quality of the model response. We treat $y$ as a latent target variable: it conceptually reflects the judgment that an ideal evaluator would assign to instance $I$. In practice, $y$ summarizes diverse aspects of evaluation—such as naturalness, persona consistency, or emotional resonance—into a single scalar score. This formulation does not assume access to a ground-truth rating for every instance; rather, our framework aims to approximate this latent distribution in a stable and reproducible manner through the use of informative auxiliary questions.

**Ground-truth Preference Distribution:** We posit the existence of an ideal but unknown distribution $p^*(y \mid I)$, which represents the rating a perfect evaluator would assign to instance $I$. While inaccessible in practice, $p^*$ serves as the conceptual target for reproducible and consistent evaluation.

**Evaluation Question:** To reduce uncertainty about $y$, the evaluator is guided by auxiliary evaluation questions $q \in Q$, each associated with a discrete answer space $\mathcal{A}_q$. Unlike free-form reasoning, these questions are designed to be short and structured, typically phrased as yes/no or small-scale categorical checks (e.g., {Yes, No} or {Poor, Fair, Good}). Thus, each answer $a \in \mathcal{A}_q$ is drawn from a limited, predefined set rather than open text. This discreteness makes it feasible to estimate the distribution $p(a \mid q, I)$ via repeated simulations, and each $(q, a)$ pair can be viewed as a partial observation that constrains the possible ratings, thereby providing additional information about the latent variable $y$.

**Objective:** Our framework seeks to construct a compact questionnaire $Q^* \subseteq Q$ that maximizes the expected information about $y$. For a question $q$ with answer $a$, the information gain is defined as

$$IG(q, a) = H[p^*(y \mid I)] - H[p^*(y \mid I \cup (q, a))], \tag{1}$$

where $H[\cdot]$ is Shannon entropy. Since $a$ is unknown beforehand, we consider the expected information gain:

$$EIG(q) = \mathbb{E}_{a \sim p(a|q,I)} \big[ IG(q,a) \big]. \tag{2}$$

Intuitively, $EIG(q)$ measures how much answering $q$ is expected to reduce our uncertainty about the final rating $y$. In other words, questions with higher EIG are the "most worth asking," since their answers lead to the largest expected reduction in rating entropy. The optimized rubric is then given by

$$Q^* = \arg \max_{Q' \subseteq Q, |Q'|=k} \sum_{q \in Q'} EIG(q). \tag{3}$$

This general formalism grounds the remainder of our methodology: candidate generation defines $Q$, simulated responses approximate $p(y, a \mid q, I)$, and the final rubric $Q^*$ emerges by optimizing for maximum expected information gain.

## 3.2 Constructing the Candidate Question Space

The first step is to construct the candidate space $Q$ of evaluation questions. The goal is not to design a perfect rubric but to provide a sufficiently diverse pool for subsequent EIG-based selection. From an information-theoretic perspective, $Q$ should approximate a high-entropy prior over evaluative dimensions, covering a wide range of response qualities while limiting redundancy. Such diversity ensures that optimization in later stages can identify the most informative subset $Q^*$. To build $Q$, we use an iterative process: the LLM is prompted with unlabeled evaluation instances to propose candidate questions (Please refer to the specific prompts in Appendix A.5), which are aggregated and deduplicated until the pool stabilizes. This process empirically converges to a stable region, confirming that the candidate pool quickly reaches sufficient diversity for subsequent selection (details in section 4.3). Importantly, our framework does not require $Q$ to be stratified or balanced across dimensions. As long as $Q$ is sufficiently varied, EIG will automatically identify and retain the most informative questions, making candidate generation a matter of maximizing breadth rather than precision.

## 3.3 Estimating the Joint Distribution of Answers and Ratings

With the candidate pool $Q$ constructed, the next step is to estimate the joint distribution $p(y, a \mid q, I)$, which underlies the computation of expected information gain (EIG). The detailed procedure is as follows.

**Discrete Answer Space:** Each evaluation question $q \in Q$ is constrained to a finite answer set $\mathcal{A}_q$ (e.g., binary, ternary, or Likert-scale). This restriction is essential: EIG requires well-defined probability distributions over answers $p(a \mid q, I)$, which are infeasible for open-ended responses. Standardizing questions into discrete formats—most often binary—keeps estimation tractable and empirically stable. We further compared binary, ternary, and 5-point scales: results show binary options achieve the most consistent evaluations, while finer granularity introduces subjectivity and drift (see Appendix A.2).

**Simulated Response Filling:** To approximate $p(a \mid q, I)$, we conduct independent experiments across $m$ distinct instances $I_j$ and questions $q$ (Please refer to the specific prompts in Appendix A.5), yielding

$$\hat{p}(a \mid q, I) = \frac{1}{m} \sum_{j=1}^{m} \mathbf{1}[a_j = a]. \tag{4}$$

Here, each $I_j$ is a different evaluation instance, and $q$ represents the same evaluation question posed to each instance. After simulating responses across $m$ samples, the information provided by our joint distribution tends to stabilize (details in section 4.3).

**Joint Distribution with Ratings:** For each sampled answer $a$, the evaluator also provides a rating $y \in \mathcal{Y}$, giving

$$\hat{p}(y, a \mid q, I) = \hat{p}(a \mid q, I) \cdot p(y \mid I, q, a). \tag{5}$$

This captures how answers correlate with final ratings.

**Outcome:** The joint distribution $\hat{p}(y, a \mid q, I)$ enables principled computation of $EIG(q)$, linking candidate generation with information-theoretic selection.

## 3.4 Expected Information Gain (EIG)

We adopt the perspective of Bayesian Experimental Design (BED): an informative question is one whose answer most reduces the uncertainty of the final rating $y$. This uncertainty is quantified by the Shannon entropy of $p(y \mid I)$, where $I$ is the evaluation instance.

**Information Gain:** For a question $q_i$ with possible answer $a_i \in \mathcal{A}_{q_i}$, the information gain is

$$IG(q_i, a_i) = H[p(y \mid I)] - H[p(y \mid I, q_i, a_i)], \tag{6}$$

measuring the reduction in entropy once $a_i$ is known.

**Expected Information Gain:** Since $a_i$ is unknown at selection time, we compute

$$EIG(q_i) = \mathbb{E}_{a_i \sim p(a_i \mid q_i, I)}\big[IG(q_i, a_i)\big], \tag{7}$$

using empirical estimates $\hat{p}$ from the simulation procedure in Section 3.3. This makes the BED criterion operational and computable.

**Optimization Objective:** The final rubric $Q*$ is obtained by selecting $k$ questions that maximize total informativeness:

$$Q^* = \arg\max_{Q' \subseteq Q, |Q'| = k} \sum_{q \in Q'} EIG(q). \tag{8}$$

This yields a compact, informative, and stable rubric, mitigating the drift of free-form judging.

## 3.5 Final Rubric Construction and Questionnaire-based Evaluation

With EIG defined, we now construct the final evaluation rubric. Given the candidate pool $Q$ and the joint-distribution estimates from Section 3.3, we compute $EIG(q)$ for each $q \in Q$ and select a compact subset that maximizes total informativeness.

**Constructing the Final Rubric:** We obtain the optimized rubric by a deterministic top-$k$ selection, following the same objective defined in Eq. 8. This procedure consolidates the outcome of prior stages into a fixed set of the most informative questions. By design, $Q^*$ balances stability (a fixed, repeatable rubric) with adaptivity (dimensions chosen to be maximally informative for the current task domain).

**Questionnaire-based Evaluation Protocol:** Once $Q^*$ is fixed, the evaluation follows a structured process where the LLM-as-a-judge assesses each question in $Q^*$. The LLM evaluates the responses and synthesizes them to provide a final score $\hat{y}$, reflecting the overall performance (Please refer to the specific prompts in Appendix A.5). Conceptually, $\hat{y}$ is a reproducible approximation to the latent rating $y$ (the judgment of an ideal evaluator). Crucially, the protocol replaces free-form, ad-hoc reasoning with a pre-committed rubric, thereby eliminating dimension drift: unlike CoT-style methods that may invent or drop dimensions across runs, our judge is anchored to the same criteria every time.

**Practical Considerations:** Based on empirical observations, we set $k = 5$ to balance evaluation cost and key dimension coverage, keeping the rubric concise yet informative. Detailed justification is provided in Appendix A.4.

## 4 Experiments

### 4.1 Experiment Setup

**Benchmarks:** We evaluate our method on six benchmarks spanning both general and domain-specific subjective tasks. CharacterEval Tu et al. (2024) measures role-playing on a 5-point Likert scale; The rJokes Weller & Seppi (2020) evaluates humor on a 10-point scale; Love-EQ, our new dataset, assesses emotional intelligence in romantic-chat scenarios on a 5-point scale with human annotations. MT_bench Zheng et al. (2023) tests general response quality with binary preferences; TSA-MD Toledo-Ronen et al. (2022) measures emotional intensity on a 5-point scale; and

Capybara-pref[1] provides large-scale response preference data on a 5-point scale. **Baselines:** We compare our EIG-based framework against four strategies: LLM Direct, which scores responses without structured reasoning; LLM + CoT, which applies Chain-of-Thought prompting for guided scoring;PoLL Verga et al. (2024), an ensemble approach that aggregates multiple LLM evaluators to improve robustness via diversity; and CheckEval Pereira et al. (2024), which operationalizes evaluation through predefined checklists of dimensions. **Metrics:** We adopt Cohen's $\kappa$ to measure reliability of repeated evaluations, defined as $\kappa = \frac{p_o - p_e}{1 - p_e}$, where $p_o$ is the observed agreement and $p_e$ the expected agreement by chance. For self-consistency, we report $\kappa_{\text{self}}$ based on Cohen's $\kappa$ across repeated LLM-as-a-judge evaluations. For human alignment, we report $p_{o\text{human}}$, the observed agreement between model-based scores and human annotations, without chance correction. Given that larger scales can lead to more discrepancies, we adjust the agreement condition accordingly: for a 5-point scale, a 1-point difference is considered consistent, and for a 10-point scale, a 2-point difference is consistent, resulting in less pronounced score differences. **Models:** We conduct main experiments on GPT-4.1 OpenAI (2025), following prior work on LLM-as-a-judge evaluation. To further test the robustness and generality of our framework across architectures and scales, we additionally evaluate with Qwen3-Next-80B-A3B-Instruct Team (2025c), Qwen3-30B-A3B-Instruct-2507 Team (2025a), DeepSeek-R1-0528 DeepSeek-AI (2025), Llama-3.3-70B-Instruct Meta (2024), and Qwen3-8B Team (2025b) (see Appendix A.6).

## 4.2 OVERALL RELIABILITY, COST EFFICIENCY, AND HUMAN ALIGNMENT

We assess three critical aspects of evaluation methods: (1) *self-consistency*, (2) *alignment with human judgments*, and (3) *cost efficiency* (measured by average LLM calls and token usage).

**Stability vs. Adaptivity (Table 1).** On one hand, the free-form **LLM + CoT** method, while flexible, demonstrates the lowest stability. Its $\kappa_{\text{self}}$ score is the worst among all baselines (average of 0.6132), reflecting the instability caused by *dimension drift*, i.e., shifting evaluation criteria across repeated trials. On the other hand, the **CheckEval** method exemplifies the opposite side of the trade-off: its fixed checklist yields higher self-consistency but the lowest $p_{o\text{human}}$, highlighting the brittleness of rigid rubrics that fail to capture task-specific nuances. In contrast, our proposed **EIG** framework achieves the highest scores on *both* $\kappa_{\text{self}}$ and $p_{o\text{human}}$. On average, it improves self-consistency by $+7.6\%$ over CoT and $+3.6\%$ over CheckEval, while also attaining the strongest alignment with human judgments. By selecting the most informative questions, our information-theoretic rubric alleviate the stability–adaptivity dilemma: it anchors evaluations to consistent criteria (mitigating dimension drift) while remaining task-adaptive.

**Cost Dimension.** While Table 1 highlights performance advantages, a key motivation for LLM-as-a-judge is to offer a scalable and cost-effective alternative to human annotation. To capture this practical dimension,in our framework, we go beyond single-run cost and introduce the notion of amortized cost, defined as Amortized Cost per Sample $= \frac{\text{Upfront Cost}}{N} +$ Per-sample Cost, where $N$ is the number of evaluated samples, *Upfront Cost* refers to the one-time expense of constructing the optimized questionnaire $Q^*$, and Per-sample Cost denotes the cost of applying it to each evaluation.

Figure 3 illustrates the *cost–ability trade-off* across methods. Here, the X-axis measures amortized cost, the Y-axis reports $\kappa_{\text{self}}$, and the dot size encodes $p_{o\text{human}}$. From this perspective, LLM + CoT incurs moderate per-sample cost but exhibits the lowest stability ($\kappa_{\text{self}}$), reflecting dimension drift. CheckEval achieves higher stability yet shows weak human alignment (small dot size). PoLL delivers robustness but its amortized cost grows linearly with $N$, making it prohibitive for large-scale evaluations. By contrast, our EIG method requires an initial one-time investment to construct $Q^*$, but once established, the amortized cost per sample decreases rapidly as $N$ grows and quickly converges to a low, stable level.

## 4.3 EFFECT OF HYPERPARAMETERS

We study the sensitivity of our framework to two key hyperparameters: (i) the number of unlabeled samples used during questionnaire generation, and (ii) the number of samples used for simulated

---

[1] https://huggingface.co/datasets/argilla/Capybara-Preferences

Table 1: Comparison of self-consistency ($\kappa_{\text{self}}$, measured by Cohen's $\kappa$) and human alignment ($p_{o\text{human}}$, measured by Observed agreement). Best results are in bold, second-best are shaded.

| Method | $\kappa_{\text{self}}$ (Cohen's $\kappa$) | | | | | | | $p_{o\text{human}}$ (Observed agreement) | | | | | | |
| --- | --- | --- | --- | --- | --- | --- | --- | --- | --- | --- | --- | --- | --- | --- |
| | CharEval | rJokes | Love-EQ | MT_bench | TSA-MD | Capybara | Avg | CharEval | rJokes | Love-EQ | MT_bench | TSA-MD | Capybara | Avg |
| LLM Direct | 0.6218 | 0.4420 | 0.7924 | 0.9314 | 0.7768 | 0.8519 | 0.7361 | 0.6534 | 0.4526 | 0.6736 | 0.5864 | 0.5949 | 0.7125 | 0.6122 |
| LLM + CoT | 0.5351 | 0.3329 | 0.6813 | 0.8494 | 0.6561 | 0.6242 | 0.6132 | 0.6974 | 0.5061 | 0.7126 | 0.6133 | 0.6238 | 0.7300 | 0.6472 |
| PoLL | 0.6317 | 0.4479 | 0.7834 | 0.9407 | 0.7714 | 0.8497 | 0.7375 | 0.6627 | 0.4417 | 0.6818 | 0.5957 | 0.6045 | 0.7025 | 0.6148 |
| CheckEval | 0.6398 | 0.4533 | 0.8143 | 0.9434 | 0.7821 | 0.8503 | 0.7472 | 0.5773 | 0.4061 | 0.5834 | 0.5127 | 0.5397 | 0.6925 | 0.5520 |
| **Ours (EIG)** | **0.6728** | **0.4832** | **0.8513** | **0.9523** | **0.8145** | **0.8974** | **0.7786** | **0.7186** | **0.5394** | **0.7455** | **0.6533** | **0.6367** | **0.7925** | **0.6810** |

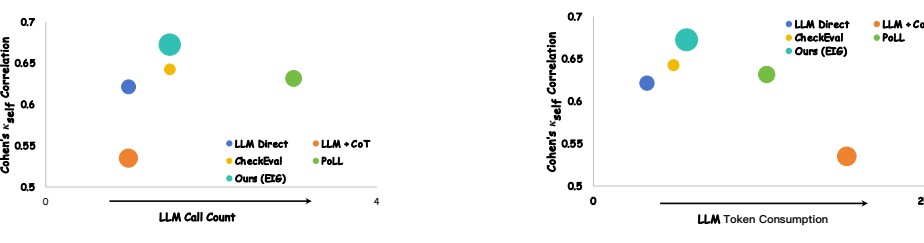

(a) Cost measured by LLM Calls      (b) Cost measured by Token Usage

Figure 3: Cost–ability trade-offs across methods. X-axis: computational cost (LLM calls or token usage). Y-axis: self-consistency ($\kappa_{\text{self}}$). Dot size: alignment with human judgments ($p_{o\text{human}}$).

response filling. For the first hyperparameter, we examine how the diversity of generated questions evolves as more unlabeled samples are referenced. Figure 4(a) shows the average across six datasets: the number of unique (deduplicated) questions grows but quickly plateaus, indicating that a sufficiently high-entropy candidate pool can be obtained with a reasonable sample size. This ensures that EIG-based selection operates over a diverse yet stable set of evaluative dimensions. For the second hyperparameter, we analyze how the stability of question selection changes with more simulated answers. Figure 4(b) illustrates one CharacterEval role: normalized EIG values gradually stabilize as the number of simulated responses increases, showing that question importance rankings converge once the sample size is large enough. This convergence anchors the final rubric to reproducible criteria rather than stochastic fluctuations. Together, these results empirically support our **stability–informativeness closed loop**: candidate diversity guarantees adaptivity, while EIG-based stabilization secures consistency, jointly suppressing the dimension drift that undermines free-form protocols such as CoT.

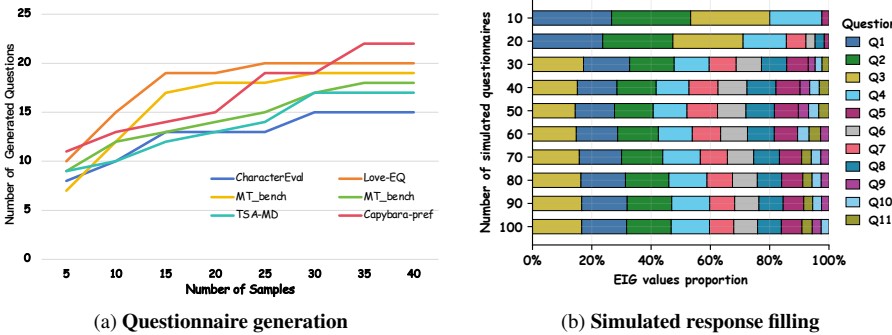

(a) **Questionnaire generation**      (b) **Simulated response filling**

Figure 4: Effect of hyperparameters. Left: question diversity plateaus with larger sampled inputs. Right: EIG rankings stabilize with more simulated responses.

### 4.4 DIMENSION-BASED EXACT MATCH (EM).

To quantitatively analyze the dimension drift in Figure 1(b), we introduce the dimension-based exact match metric ($\text{EM}_{\text{dim}}$). An LLM first extracts a set of evaluation dimensions $G = \{g_1, g_2, \ldots, g_m\}$ from the initial subjective evaluation. Then $G$ and the re-test evaluation are provided to the LLM, which outputs the proportion of dimensions in $G$ preserved in the second response.

Formally, $\mathrm{EM}_{\mathrm{dim}} = k/|G|$, where $|G|$ is the number of extracted dimensions and $k$ the subset preserved in the second response. This measures how well initial evaluation dimensions are retained in retesting. As shown in Figure 5, LLM + CoT analyses exhibit mild drift on simpler tasks (e.g., **MT_bench**, **The rJokes**) but more pronounced drift on complex ones (e.g., **CharacterEval**, **TSA-MD**). By contrast, EIG achieves $\mathrm{EM}_{\mathrm{dim}} = 1.0$ across all tasks since the fixed questionnaire $Q^*$ eliminates drift. Importantly, Table 1 shows that EIG also reaches the highest human alignment ($p_{o\mathrm{human}}$). Thus, unlike rigid checklist baselines such as CheckEval, EIG combines *stability* (no drift) with *adaptivity* (task-relevant dimensions), alleviating the stability–adaptivity dilemma discussed in the introduction.

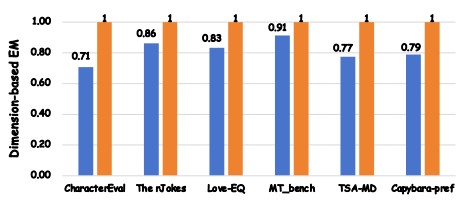

Figure 5: Dimension drift measured by $\mathrm{EM}_{\mathrm{dim}}$. LLM + CoT shows mild drift on simpler tasks (e.g., MT_bench) and much stronger drift on complex ones (e.g., CharacterEval).

Finally, $\mathrm{EM}_{\mathrm{dim}}$ should be understood as an approximate measure whose reliability depends on the LLM. Its value lies not in providing absolute statistics, but in offering a quantitative perspective that captures *relative trends* of dimension drift across methods.

### 4.5 ROBUSTNESS EXPERIMENT

We examine the robustness of our approach under different conditions. Figure 6a reports $\kappa_{\mathrm{self}}$ and $p_{o\mathrm{human}}$ for different foundation models on the rJokes dataset. For Qwen3, $\kappa_{\mathrm{self}}$ decreases as the model size increases, while $p_{o\mathrm{human}}$ improves. This may result from larger models exhibiting both more divergent thinking and more reliable reasoning. Similar patterns are observed for DeepSeek-R1, Llama3.3, and GPT-4.1, indicating that the phenomenon is not model-specific.

Figure 6b compares the final questionnaire under three strategies: "All candidate questions," "Random k questions," and "Ours." Both baselines underperform our method, but their behaviors differ. "All" achieves higher $p_{o\mathrm{human}}$, suggesting stronger alignment with human judgment, whereas "Random" obtains higher $\kappa_{\mathrm{self}}$, reflecting stronger internal consistency. Our approach balances both metrics and consistently outperforms.

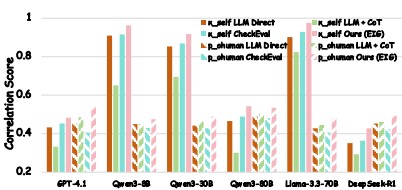

(a) Different foundation models' performance

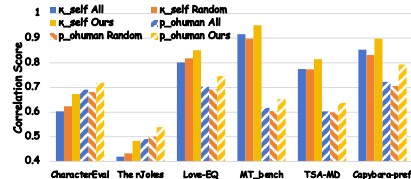

(b) Performance of different final questionnaire selection methods

Figure 6: Robustness analysis. Left: $\kappa_{\mathrm{self}}$ and $p_{o\mathrm{human}}$ across models on The rJokes. Right: comparison of questionnaire selection strategies.

## 5 CONCLUSION

We identified dimension drift as a core obstacle to reproducible LLM-based subjective evaluation and proposed an EIG-driven rubric framework to resolve it. By selecting the most informative evaluation questions, our method achieves both stability and adaptivity in a unified design. Experiments across six benchmarks show consistent gains in self-consistency, human alignment, and cost efficiency over existing baselines. This establishes information-theoretic rubric construction as a principled foundation for scalable and trustworthy LLM-as-a-judge systems.

LIMITATIONS

While our study demonstrates that EIG achieves both stability and adaptivity with favorable long-term cost efficiency, several limitations remain. First, our dimension-based exact match metric ($EM_{dim}$) is only an approximate measure whose reliability depends on the underlying LLM. Second, although the amortized cost of EIG is low in large-scale settings, its upfront investment may limit applicability in domains that change rapidly, requiring new questionnaires. Finally, we did not explicitly investigate robustness under adversarial prompts or strong domain shifts. We leave these directions to future work.

ETHICS STATEMENT

This research adheres to the ICLR Code of Ethics. We ensure ethical conduct throughout the study, with no human subjects involved. Ethical considerations, such as fairness, potential biases, and privacy, have been addressed. Our framework is designed for using LLMs as evaluators for subjective tasks, ensuring that the evaluation results are more consistent and reliable.

THE USE OF LARGE LANGUAGE MODELS

Under the policy of ICLR, a large language model was used solely to polish the language of the manuscript, improving grammar, fluency, and clarity. It did not assist in scientific reasoning, or contribute to technical content. All ideas, methods, experiments, and conclusions are the original work of the authors. The final manuscript was thoroughly reviewed, verified, and approved by all human authors to ensure scientific accuracy and integrity.

REPRODUCIBILITY STATEMENT

We are committed to reproducibility. The source code and datasets used in the experiments is available at the following anonymous repository for reproducibility purposes: https://anonymous.4open.science/r/EIG-LLM-Eval-13FD/. Detailed descriptions of the methodology, including the construction of rubrics using Expected Information Gain (EIG) and the experimental setup, are provided. All steps taken to ensure reproducibility, including the use of multiple LLMs and precise experimental parameters, are documented for verification.

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

# A APPENDIX

## A.1 THE USE OF LARGE LANGUAGE MODELS (LLMS)

We utilized Large Language Models (LLMs) to check for language issues and enhance the clarity and fluency of the paper.

## A.2 EFFECT OF RESPONSE GRANULARITY ON EIG STABILITY

Section 1 identified dimension drift as the main source of instability in free-form evaluation. Our EIG framework mitigates this by fixing a questionnaire, suppressing macro-level drift. A natural question is whether the answer format itself introduces instability at a finer scale. We test this on CharacterEval by holding the question set fixed and varying answer granularity: binary (yes/no), ternary, and 5-point Likert (Fig. 7). Results show a clear trade-off: (i) with more options, absolute EIG differences shrink; (ii) however, question rankings by EIG become less stable, as finer-grained scales amplify LLM subjectivity; (iii) binary options yield the most consistent rankings. Thus, response granularity introduces a form of micro-level drift. To build a robust evaluation system, both informative question selection and stable answer formats are necessary. Binary choices, beyond computational simplicity, are a key design choice for ensuring end-to-end stability.

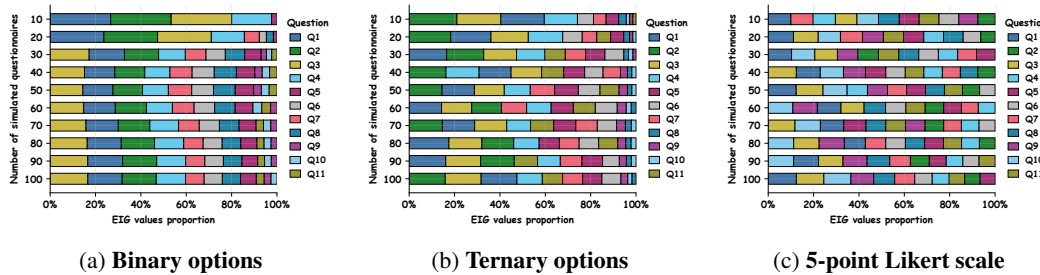

(a) **Binary options**      (b) **Ternary options**      (c) **5-point Likert scale**

Figure 7: **Effect of response granularity on EIG.** Simulated responses with increasing sample size show how expected information gain (EIG) changes under binary, ternary, and 5-point Likert settings.

### A.3 Algorithmic Details of Questionnaire Construction and Evaluation

To ensure transparency and reproducibility, we provide the full pseudocode for our Expected Information Gain (EIG)-based rubric construction. This algorithm instantiates the Bayesian Experimental Design perspective described in Section 3, where evaluation questions are treated as experiments whose utility is measured by the expected reduction in uncertainty about the latent rating.

The procedure begins by generating a diverse pool of candidate evaluation questions $Q$ using LLM proposals on unlabeled instances. Each question is standardized into a discrete answer space $\mathcal{A}_q$ (e.g., binary or small categorical options), ensuring tractable probability estimation and eliminating duplicates. For each question–instance pair, we approximate the answer distribution $\hat{p}(a \mid q, I)$ via repeated LLM simulations, and conditionally elicit corresponding ratings $p(y \mid I, q, a)$. Together, these yield an empirical joint distribution $\hat{p}(y, a \mid q, I)$, which anchors the subsequent information-theoretic computation.

We then compute the Expected Information Gain (EIG) of each candidate question:

$$EIG(q) = \frac{1}{|\mathcal{D}|} \sum_{I \in \mathcal{D}} \sum_{a \in \mathcal{A}_q} \hat{p}(a \mid q, I) \Big( H[p(y \mid I)] - H[p(y \mid I, q, a)] \Big).$$

Intuitively, questions with higher EIG values contribute more to reducing entropy in the rating distribution, and are thus more informative for stabilizing evaluation. The final rubric $Q^*$ is obtained via a top-$k$ selection that maximizes the total informativeness across all chosen questions.

During evaluation, the LLM-as-a-judge is constrained to answer only the rubric questions in $Q^*$, producing structured answers $\{a_q\}_{q \in Q^*}$. These answers are then aggregated into a reproducible rating $\hat{y}$ through a deterministic mapping, ensuring stability and eliminating dimension drift.

The complete procedure is summarized in Algorithm 1.

This pseudocode operationalizes the stability–informativeness loop described in the main text: candidate diversity enables adaptivity, EIG quantifies informativeness, and the final rubric enforces stability across repeated evaluations.

### A.4 Number of Questions k

Based on experiments conducted on six datasets, the **bars** in the figure represent the average Cohen's $\kappa_{\text{self}}$ correlation scores, while the **line** represents the average $p_{o\text{human}}$ Observed agreement scores. It can be observed that $\kappa_{\text{self}}$ remains relatively stable as the number of questions $k$ increases, but starts to decline when $k > 5$. In contrast, $p_{o\text{human}}$ first increases with larger $k$, and then gradually decreases. Therefore, we set $k = 5$ as the parameter for our main experiments.

### A.5 Prompt Template for Questionnaire Generation

In this subsection, we present the prompt templates used for generating candidate questions, simulating questionnaire responses and final evaluation.

---

**Algorithm 1** The Detailed Procedures of Questionnaire Construction and Evaluation

---

**Require:** Evaluation instances $\mathcal{D} = \{I = (S, M)\}$, rating space $\mathcal{Y}$, simulation rounds $m$, target size $k$

**Ensure:** Final rubric $Q^*$ and reproducible rating $\hat{y}$

1: Generate candidate questions $Q$ via LLM proposals on $\mathcal{D}$
2: Standardize each $q \in Q$ into a discrete answer space $\mathcal{A}_q$ and remove duplicates
3: **for** each $q \in Q$ and $I \in \mathcal{D}$ **do**
4:      Estimate $\hat{p}(a \mid q, I)$ by $m$-round simulation
5:      **for** each $a \in \mathcal{A}_q$ **do**
6:          Obtain conditional distribution $p(y \mid I, q, a)$
7:          Compute $\hat{p}(y, a \mid q, I) = \hat{p}(a \mid q, I) \cdot p(y \mid I, q, a)$
8:      **end for**
9: **end for**
10: **for** each $q \in Q$ **do**
11:      Compute $EIG(q) = \frac{1}{|\mathcal{D}|} \sum_{I \in \mathcal{D}} \sum_{a \in \mathcal{A}_q} \hat{p}(a \mid q, I)\big(H[p(y \mid I)] - H[p(y \mid I, q, a)]\big)$
12: **end for**
13: Select final rubric $Q^* = \arg\max_{Q' \subseteq Q, |Q'| = k} \sum_{q \in Q'} EIG(q)$
14: For evaluation: ask LLM-as-a-judge each $q \in Q^*$ on instance $I$
15: Aggregate answers $\{a_q\}_{q \in Q^*}$ into final score $\hat{y}$
        **return** $Q^*, \hat{y}$

---

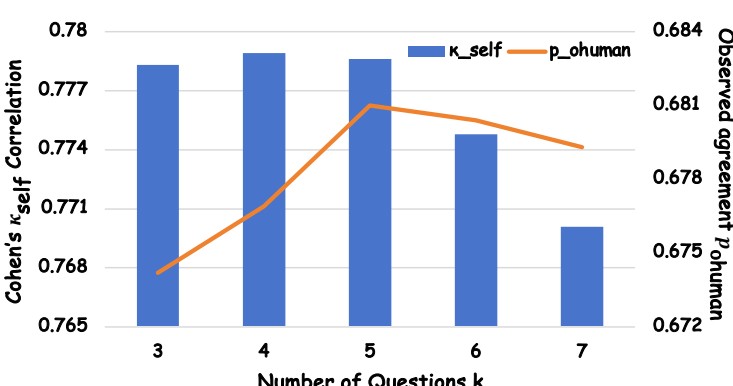

Figure 8: Results of LLM Direct, LLM CoT, and Ours on human consistency metrics

### A.6 THE PERFORMANCE OF DIFFERENT FOUNDATION MODELS

Table 2: Qwen3-8B Comparison of self-consistency ($\kappa_{\text{self}}$, measured by Cohen's $\kappa$) and human alignment ($p_{o\text{human}}$, measured by Observed agreement). Best results are in bold, second-best are shaded.

| Method | $\kappa_{\text{self}}$ (Cohen's $\kappa$) | | | | | | $p_{o\text{human}}$ (Observed agreement) | | | | | |
|---|---|---|---|---|---|---|---|---|---|---|---|---|
| | CharEval | rJokes | Love-EQ | MT_bench | TSA-MD | Capybara | CharEval | rJokes | Love-EQ | MT_bench | TSA-MD | Capybara |
| LLM Direct | 0.7275 | 0.9392 | 0.8512 | 0.9167 | 0.8585 | 0.9283 | 0.5172 | 0.4606 | 0.6451 | 0.5742 | 0.5158 | 0.6350 |
| LLM + CoT | 0.6709 | 0.6501 | 0.7048 | 0.8685 | 0.7449 | 0.8127 | 0.4934 | 0.4667 | 0.6126 | 0.5600 | 0.5507 | 0.6734 |
| CheckEval | 0.7245 | 0.9261 | 0.8783 | 0.8439 | 0.8923 | 0.8945 | 0.3840 | 0.4302 | 0.5836 | 0.4767 | 0.4841 | 0.6097 |
| **Ours (EIG)** | **0.7545** | **0.9622** | **0.8941** | **0.9831** | **0.9203** | **0.9694** | **0.5808** | 0.4758 | **0.7124** | **0.6033** | **0.5916** | **0.6847** |

### A.7 LOVE-EQ ANNOTATION GUIDELINES AND PROCEDURE

#### A.7.1 ANNOTATION GUIDELINE

This guideline was provided to all annotators participating in the construction of the Love-EQ dataset. The annotation aims to assess the overall dialogue quality and emotional intelligence (EQ) exhibited by the *male speaker* in romantic/flirting chat scenarios. Specifically, for each dialogue,

---

## Prompt template for Candidate Generation

### System Prompt

As a role-playing dialogue evaluation expert, please generate a series of "Yes/No" format evaluation questions based on the provided character information, evaluation dimensions, and reference samples. The aim is to comprehensively assess the dialogue quality of the character.

- - - - - - - - - - - - - - - - - - - - - - - - - - - - - - - - - - - - - - - - - - - - - -

### Instruction

**<Role Profile>**: 'Name': Tong Xiangyu; 'Identity': Owner of Tongfu Inn, daughter of the Longmen Escort Agency; 'Personality': Selfish, greedy, vain, and concerned with appearances, yet also seeks enjoyment; 'Background': Originally set to marry the head of Mount Heng Sect, she became a widow on the way. She ...

**<Reference Samples>**: Tong Xiangyu: (playfully whining) "Oh, come on! Such a simple question, and you need so long to think?"

Bai Zhantang: "Can we change the question?"

…

Tong Xiangyu: "Hmph, so funny. If you have such a great sense of humor, why don't you go write a sitcom?"

Bai Zhantang: "I still prefer careers with ambition, like being a waiter."

**<Task Requirements>:** Please generate a series of "Yes/No" evaluation questions based on the specific information and evaluation dimensions for the character. The specific requirements are as follows:

1. Each question should be in the "Yes/No" format, with answers being "Yes" or "No."
2. All questions should cover various aspects of the evaluation dimensions and be designed based on the character's specific traits and background.
3. The questions should be clear, specific, and objective, making it easy for evaluators to make judgments.
4. The question design should closely align with the character's traits, reflecting their performance in dialogue.
5. Each question should focus on one evaluation point, avoiding lengthy or multi-part questions.

Please output the list of questions directly, one question per line.

Figure 9: Prompt for Generating Candidate Questions

Table 3: Qwen3-30B-A3B-Instruct-2507 Comparison of self-consistency ($\kappa_{\text{self}}$, measured by Cohen's $\kappa$) and human alignment ($p_{o\text{human}}$, measured by Observed agreement). Best results are in bold, second-best are shaded.

| Method | $\kappa_{\text{self}}$ (Cohen's $\kappa$) | | | | | | $p_{o\text{human}}$ (Observed agreement) | | | | | |
|---|---|---|---|---|---|---|---|---|---|---|---|---|
| | **CharEval** | **rJokes** | **Love-EQ** | **MT_bench** | **TSA-MD** | **Capybara** | **CharEval** | **rJokes** | **Love-EQ** | **MT_bench** | **TSA-MD** | **Capybara** |
| LLM Direct | 0.7327 | 0.9237 | 0.7734 | 0.9028 | 0.8325 | 0.9024 | 0.5124 | 0.4421 | 0.5824 | 0.5545 | 0.5347 | 0.6034 |
| LLM + CoT | 0.6342 | 0.7241 | 0.6823 | 0.8482 | 0.7461 | 0.8427 | 0.5471 | 0.4628 | 0.6323 | 0.5943 | 0.5746 | 0.6456 |
| CheckEval | 0.7417 | 0.9382 | 0.7981 | 0.9313 | 0.8512 | 0.9142 | 0.4934 | 0.4273 | 0.5593 | 0.5435 | 0.5245 | 0.5841 |
| **Ours (EIG)** | **0.7823** | **0.9672** | **0.8455** | **0.9855** | **0.8835** | **0.9575** | **0.5832** | **0.4893** | **0.6951** | **0.6248** | **0.6035** | **0.6728** |

annotators are asked to assign a **single holistic score** on a **5-point scale (1–5)** to the male utterance marked as **"Golden"**, *after considering the full conversational context*.

- **Task**: Given a multi-turn romantic/flirting conversation (with both male and female utterances), annotators should **focus only on the male "Golden" utterance**. Using the full context, annotators judge whether the male speaker's response is natural, appropriate, humorous/clever when applicable, and emotionally attuned. Annotators do *not* score each dimension separately; instead, they should weigh the following dimensions as references and finally provide one overall quality score (1–5).

- **Reference dimensions (for guidance only; no per-dimension scoring required)**:

---

**Prompt template for Simulated Response Filling**

**System Prompt**

As a role-playing dialogue evaluation expert, you will conduct a detailed assessment of the provided role-playing dialogue sample. Please answer each question based on the content of the dialogue, the character profile, and the evaluation questionnaire, and provide an overall score based on your answers.

- - - - - - - - - - - - - - - - - - - - - - - - - - - - - - - - - - - - - - - - - - - - - - - -

**Instruction**

**<Role Profile>**: 'Name': Tong Xiangyu;     'Identity': Owner of Tongfu Inn, daughter of the Longmen Escort Agency;     'Personality': Selfish, greedy, vain, and concerned with appearances, yet also seeks enjoyment;     'Background': Originally set to marry the head of Mount Heng Sect, she became a widow on the way. She ...

**<Reference Samples>**: Tong Xiangyu: (playfully whining) "Oh, come on! Such a simple question, and you need so long to think?"

Bai Zhantang: "Can we change the question?"

…

Tong Xiangyu: "Hmph, so funny. If you have such a great sense of humor, why don't you go write a sitcom?"

Bai Zhantang: "I still prefer careers with ambition, like being a waiter."

**<Task Requirements>**:

1. Please answer each question objectively and accurately, based on the character profile and the content of the dialogue.

2. For each question, pay special attention to whether the character matches the traits as defined in their profile and how they are portrayed in the dialogue.

3. The final score should consider the performance across all evaluation questions in the dialogue.

Please strictly follow the format below for your response:

Question 1: Yes/No

...

Question {len(questions)}: Yes/No

Overall score: X points

---

Figure 10: Prompt for Simulating Questionnaire Responses

Table 4: Qwen3-Next-80B-A3B-Instruct Comparison of self-consistency ($\kappa_{\text{self}}$, measured by Cohen's $\kappa$) and human alignment ($p_{o\text{human}}$, measured by Observed agreement). Best results are in bold, second-best are shaded.

| Method | $\kappa_{\text{self}}$ (Cohen's $\kappa$) | | | | | | $p_{o\text{human}}$ (Observed agreement) | | | | | |
| | CharEval | rJokes | Love-EQ | MT.bench | TSA-MD | Capybara | CharEval | rJokes | Love-EQ | MT.bench | TSA-MD | Capybara |
|---|---|---|---|---|---|---|---|---|---|---|---|---|
| LLM Direct | 0.5509 | 0.4661 | 0.6942 | 0.8437 | 0.6423 | 0.6258 | 0.6841 | 0.4892 | 0.6507 | 0.6024 | 0.6248 | 0.7267 |
| LLM + CoT | 0.4638 | 0.3016 | 0.6137 | 0.8024 | 0.6126 | 0.5742 | 0.7213 | 0.5031 | 0.6751 | 0.6303 | 0.6651 | 0.7543 |
| CheckEval | 0.5713 | 0.4892 | 0.7236 | 0.8591 | 0.6637 | 0.6484 | 0.6529 | 0.4819 | 0.6044 | 0.5592 | 0.6034 | 0.6845 |
| **Ours (EIG)** | **0.6092** | **0.5426** | **0.7949** | **0.8963** | **0.7024** | **0.6852** | **0.7587** | **0.5344** | **0.6902** | **0.6894** | **0.6892** | **0.7884** |

  – **Naturalness and fluency**: Is the wording natural, fluent, conversational, and easy to understand, without obvious "machine-like" stiffness?

  – **Emotional attunement**: Does the reply recognize and respond to the interlocutor's emotions and implicit signals, showing care, interest, or emotional alignment?

  – **Creativity and humor**: Does the reply show creativity, wit, or humor (when appropriate), rather than being mechanical, dull, or completely off-topic?

  – **Politeness and respect**: Even in teasing/flirting contexts, does the reply maintain basic respect and boundaries, avoiding clear offense or insult?

  – **Context understanding**: Is the reply coherent and consistent with the prior turns, avoiding obvious non sequiturs or misunderstandings of the situation?

```
Prompt template for Final Evaluation

System Prompt
As a role-playing dialogue evaluation expert, you will conduct a detailed assessment of the
provided role-playing dialogue sample. Please answer each question based on the content of
the dialogue, the character profile, and the evaluation questionnaire, and provide an overall
score based on your answers.
- - - - - - - - - - - - - - - - - - - - - - - - - - - - - - - - - - - - - - - - - - - - -

Instruction
<Role Profile>: 'Name': Tong Xiangyu;        'Identity': Owner of Tongfu Inn, daughter of the
Longmen Escort Agency;        'Personality': Selfish, greedy, vain ...
<Evaluated Samples>: Tong Xiangyu: (playfully whining) "Oh, come on! Such a simple
question, and you need so long to think?"
Bai Zhantang: "Can we change the question?"
…

<Evaluation Questionnaire>:
1. Does the character's response reflect their personality traits (e.g., quirky, lively,
straightforward, etc.)?
2. Does the speaking style align with their historical or martial arts background and life
experiences?
…

<Task Requirements>:
1. Base your responses strictly on the evaluation questions from the questionnaire; do not
provide any additional reasoning or explanations.
2. Maintain objectivity and neutrality, without any subjective opinions.
3. Only output the numeric score, without including any other form of response.
4. Each question should be answered based on the character's traits and background,
strictly evaluating their performance in dialogue.

Respond in this format:
**Score**: [1-5]
```

Figure 11: Prompt for Simulating Questionnaire Responses

Table 5: Llama-3.3-70B-Instruct Comparison of self-consistency ($\kappa_{\text{self}}$, measured by Cohen's $\kappa$) and human alignment ($p_{o\text{human}}$, measured by Observed agreement). Best results are in bold, second-best are shaded.

| Method | $\kappa_{\text{self}}$ (Cohen's $\kappa$) | | | | | | $p_{o\text{human}}$ (Observed agreement) | | | | | |
|---|---|---|---|---|---|---|---|---|---|---|---|---|
| | CharEval | rJokes | Love-EQ | MT_bench | TSA-MD | Capybara | CharEval | rJokes | Love-EQ | MT_bench | TSA-MD | Capybara |
| LLM Direct | 0.8127 | 0.9212 | 0.8542 | 0.9633 | 0.8272 | 0.9242 | 0.4671 | 0.4273 | 0.6118 | 0.5667 | 0.5237 | 0.5875 |
| LLM + CoT | 0.7904 | 0.8230 | 0.7951 | 0.9230 | 0.7731 | 0.8729 | 0.5026 | 0.4455 | 0.6572 | 0.5800 | 0.5648 | 0.6383 |
| CheckEval | 0.8433 | 0.9588 | 0.8626 | 0.9845 | 0.8453 | 0.9360 | 0.4248 | 0.4061 | 0.5642 | 0.5667 | 0.5039 | 0.5240 |
| Ours (EIG) | **0.8702** | **0.9755** | **0.8873** | **1.0000** | **0.8946** | **0.9607** | **0.5392** | **0.4818** | **0.6837** | **0.6176** | **0.5902** | **0.6639** |

- **Holistic rating rubric (1–5)**:

    - **5 – Excellent**: Strong overall performance; highly aligned with romantic chat contexts and outstanding on most reference dimensions.

    - **4 – Good**: Generally good with minor flaws; overall natural, appropriate, and reasonably engaging.

    - **3 – Neutral/Average**: Mixed strengths and weaknesses; acceptable but unremarkable, or slightly problematic on some dimensions.

    - **2 – Poor**: Multiple noticeable issues (e.g., unnatural phrasing, weak empathy, lack of humor/creativity, or mildly disrespectful behavior).

Table 6: DeepSeek-R1-0528 Comparison of self-consistency ($\kappa_{\text{self}}$, measured by Cohen's $\kappa$) and human alignment ($p_{o\text{human}}$, measured by Observed agreement). Best results are in bold, second-best are shaded.

| Method | $\kappa_{\text{self}}$ (Cohen's $\kappa$) | | | | | | $p_{o\text{human}}$ (Observed agreement) | | | | | |
| --- | --- | --- | --- | --- | --- | --- | --- | --- | --- | --- | --- | --- |
| | CharEval | rJokes | Love-EQ | MT_bench | TSA-MD | Capybara | CharEval | rJokes | Love-EQ | MT_bench | TSA-MD | Capybara |
| LLM Direct | 0.5724 | 0.3706 | 0.7451 | 0.9038 | 0.6829 | 0.6347 | 0.6719 | 0.4532 | 0.6951 | 0.6467 | 0.6127 | 0.7346 |
| LLM + CoT | 0.5247 | 0.2937 | 0.7193 | 0.8450 | 0.6351 | 0.6039 | 0.7135 | 0.4609 | 0.6845 | 0.6533 | 0.6412 | 0.7639 |
| CheckEval | 0.5912 | 0.3934 | 0.7517 | 0.9126 | 0.6933 | 0.6732 | 0.6028 | 0.4242 | 0.5927 | 0.6128 | 0.5892 | 0.7014 |
| **Ours (EIG)** | **0.6138** | **0.4288** | **0.7958** | **0.9364** | **0.7453** | **0.7192** | **0.7462** | **0.4939** | **0.7383** | **0.6769** | **0.6553** | **0.7933** |

  – **1 – Very poor**: Largely unacceptable (e.g., severely unnatural, strongly offensive, severely misunderstanding the context, or offering almost no positive interaction value).

- **Reference materials and ambiguity handling**: Annotators are provided with (i) the full multi-turn dialogue, (ii) a brief scenario description, and (iii) standardized definitions and examples for each reference dimension. For ambiguous cases (e.g., seemingly rude wording used as friendly banter), annotators should judge based on the **overall conversational tone and the interlocutor's reaction**, reflecting realistic flirting/teasing interactions.

Before formal annotation, all annotators completed a **calibration phase**: they rated a shared set of examples and compared outcomes against reference ratings and group discussion. Only after achieving sufficient alignment did annotators proceed to the main annotation stage.

### A.7.2 ANNOTATION PROCEDURE

Following the guideline above, we conducted systematic annotation for Love-EQ. We first curated multi-turn dialogue instances from cleaned and de-identified romantic chat data, retaining samples that contain a male **"Golden"** utterance, and then applied unified formatting and noise filtering. **The "Golden" markers were initially assigned by GPT-4.1 and subsequently reviewed and filtered by human annotators** to ensure correctness and relevance. We recruited annotators with Chinese reading proficiency and familiarity with online chat conventions, and provided standardized training and trial annotation to ensure a consistent understanding of the task and reference dimensions.

In the main annotation stage, each instance (i.e., one male Golden utterance with its full context) was independently rated by **three** annotators, each assigning a **single holistic score (1–5)** while freely weighing the reference dimensions. The final dataset contains **1,039** annotated instances. To ensure quality, we inserted a small set of validation items with relatively unambiguous expected ratings, and manually reviewed conspicuous disagreements or abnormal annotations for re-checking and potential re-annotation.

After annotation, we quantified inter-annotator agreement on the 1–5 holistic ratings using **Fleiss' $\kappa$**, computed over the three annotators' ratings, obtaining $\kappa = 0.7326$.

### A.8 SAMPLING STRATEGIES FOR SIMULATED CONTEXTS IN EIG ESTIMATION

To estimate $p(a \mid q, I)$ with $m$ simulated contexts, our main pipeline constructs each context $I$ by **uniformly sampling** instances from the dataset $D$ at random. This simple strategy serves as a practical approximation to the overall data distribution and is used throughout the main experiments.

To examine whether the EIG-based question ranking is sensitive to how simulated contexts are sampled, we additionally compare the above random sampling against a **cluster-then-stratified sampling** strategy. Concretely, we first cluster all instances in $D$ (e.g., using text representations/embeddings) into multiple clusters, and then construct each simulated context by sampling instances in a **stratified** manner across clusters (e.g., proportional allocation or fixed quota per cluster). Intuitively, this variant encourages broader coverage of diverse conversational patterns within each simulated context.

Figure 12 shows the **converged** EIG-induced question ranking under the two sampling schemes. As can be seen, the resulting rankings are **highly consistent** after convergence, with only minor local swaps among adjacent questions. Therefore, we adopt the simpler **uniform random sampling** method in the main pipeline.

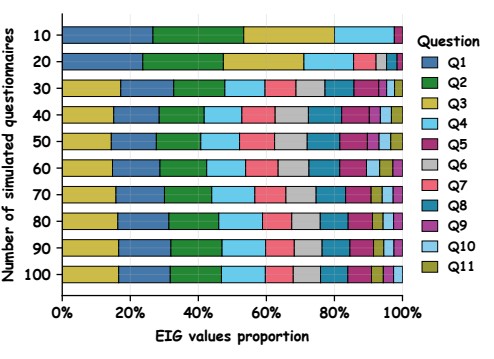
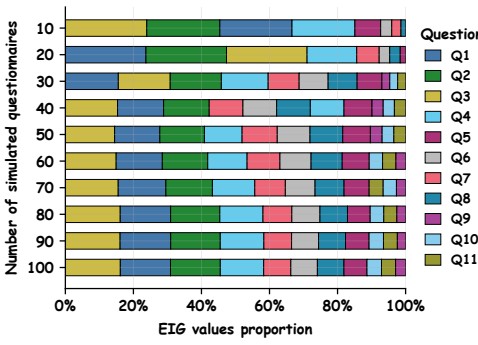

(a) Uniform random sampling from $D$.  (b) Cluster-then-stratified sampling.

Figure 12: Comparison of converged EIG-based question rankings under different strategies for sampling $m$ simulated contexts when estimating $p(a \mid q, I)$. The rankings are highly consistent across sampling schemes, supporting the use of uniform random sampling in the main experiments.

Table 7: Comparison of self-consistency ($\kappa_{\text{self}}$, measured by Cohen's $\kappa$) and human alignment ($p_{o\text{human}}$, measured by observed agreement). Best results are in bold, second-best are shaded.

| Method | $\kappa_{\text{self}}$ (Cohen's $\kappa$) | | | | | | | $p_{o\text{human}}$ (Observed agreement) | | | | | | |
|---|---|---|---|---|---|---|---|---|---|---|---|---|---|---|
| | CharEval | rJokes | Love_EQ | MT_bench | TSA_MD | Capybara | Avg | CharEval | rJokes | Love_EQ | MT_bench | TSA_MD | Capybara | Avg |
| LLM Direct | 0.6218 | 0.4420 | 0.7924 | 0.9314 | 0.7768 | 0.8519 | 0.7361 | 0.6534 | 0.4526 | 0.6736 | 0.5864 | 0.5949 | 0.7125 | 0.6122 |
| PoLL (different) | 0.6317 | 0.4479 | 0.7834 | 0.9407 | 0.7714 | 0.8497 | 0.7375 | 0.6627 | 0.4417 | 0.6818 | 0.5957 | 0.6045 | 0.7025 | 0.6148 |
| PoLL (same) | 0.6264 | 0.4457 | 0.7947 | 0.9346 | 0.7790 | 0.8528 | 0.7389 | 0.6541 | 0.4532 | 0.6783 | 0.5869 | 0.5977 | 0.7142 | 0.6141 |
| **Ours (EIG)** | **0.6728** | **0.4832** | **0.8513** | **0.9523** | **0.8145** | **0.8974** | **0.7786** | **0.7186** | **0.5394** | **0.7455** | **0.6533** | **0.6367** | **0.7925** | **0.6810** |

### A.9  ADDITIONAL BASELINE: MAJORITY VOTING WITH THE SAME JUDGE MODEL (POLL SAME)

In addition to **PoLL (different)** reported in our main experiments (majority voting across diverse model families), we further evaluate **PoLL (same)**, where we sample multiple independent judging runs from the *same* model and aggregate the final decision via majority voting. As shown in Table 7, the effect on $\kappa_{\text{self}}$ is *dataset-dependent*: PoLL (different) achieves higher self-consistency on CHAREVAL, RJOKES, and MT_BENCH, while PoLL (same) is slightly higher on LOVE_EQ, TSA_MD, and CAPYBARA. Despite these modest gains in self-consistency, both voting schemes yield only limited improvements in human alignment ($p_{o\text{human}}$) compared to **LLM Direct**. In contrast, our **EIG** method consistently delivers substantially better human alignment across all benchmarks.

### A.10  VALIDATING ROBUSTNESS TO POTENTIAL CLOSED-LOOP BIAS VIA HETEROGENEOUS CROSS-MODEL EXPERIMENTS

To examine whether our EIG-based framework is susceptible to closed-loop bias (i.e., the same or highly similar LLMs being used for both EIG estimation and evaluation), we conduct a comprehensive **cross-model validation** using a **heterogeneous pipeline**. Concretely, we decouple the two roles in our pipeline: (*i*) the *Question Generator* (EIG estimator) that produces high-information questions, and (*ii*) the *Evaluator* that performs questionnaire-style evaluation. We then permute these roles across **six distinct models** spanning different capabilities and architectures: GPT-4.1 (G); Llama-3.3-70B (L); Qwen3-8B/30B/Next-80B (Q8, Q30, Q80); and DeepSeek-R1 (D).

The results in Table 8 consistently indicate that our method captures **task-intrinsic evaluation dimensions** rather than model-specific preferences. Across heterogeneous configurations, both **human alignment** ($\rho_{\text{human}}$) and **self-consistency** ($\kappa_{\text{self}}$) remain highly stable relative to homogeneous baselines. For example, with GPT-4.1 as the evaluator, the homogeneous setting (G+G) achieves $\rho_{\text{human}}$ Avg $= 0.6810$ and $\kappa_{\text{self}}$ Avg $= 0.7786$, while replacing the question generator with Qwen3-Next-80B (Q80+G) yields nearly identical performance ($\rho_{\text{human}}$ Avg $= 0.6806$, $\kappa_{\text{self}}$ Avg $= 0.7725$). Moreover, heterogeneous pairings can even outperform homogeneous ones: under Qwen3-Next-80B evaluation, using Llama-3.3 as the question generator (L+Q80) attains higher human alignment

(Avg $= 0.6937$) than the homogeneous Q80+Q80 baseline (Avg $= 0.6917$). Overall, these findings support the universality and transferability of EIG-selected criteria across model families.

Table 8: Heterogeneous-variant experiments using different LLMs in the two stages of our framework—EIG estimation and questionnaire-based evaluation—comparing self-consistency ($\kappa_{\text{self}}$, measured by Cohen's $\kappa$) and human alignment ($p_{o\text{human}}$, measured by observed agreement). Best results are in bold, second-best are shaded.

| Method | $\kappa_{\text{self}}$ (Cohen's $\kappa$) | | | | | | | $p_{o\text{human}}$ (Observed agreement) | | | | | | |
| | CharEval | rJokes | Love-EQ | MT_bench | TSA-MD | Capybara | Avg | CharEval | rJokes | Love-EQ | MT_bench | TSA-MD | Capybara | Avg |
|---|---|---|---|---|---|---|---|---|---|---|---|---|---|---|
| G (baseline best) | 0.6398 | 0.4533 | 0.8143 | 0.9434 | 0.7821 | 0.8519 | 0.7472 | 0.6974 | 0.5061 | 0.7126 | 0.6133 | 0.6238 | 0.7300 | 0.6472 |
| G + G | 0.6728 | **0.4832** | 0.8513 | 0.9523 | **0.8145** | **0.8974** | 0.7786 | **0.7186** | 0.5394 | 0.7455 | 0.6533 | 0.6367 | 0.7925 | **0.6810** |
| Q8 + G | 0.6642 | 0.4817 | **0.8561** | 0.9573 | 0.8121 | 0.8836 | 0.7758 | 0.7103 | 0.5346 | 0.7438 | 0.6512 | 0.6352 | **0.8025** | 0.6796 |
| Q30 + G | **0.6761** | 0.4803 | 0.8539 | **0.9623** | 0.8127 | 0.8971 | **0.7804** | 0.7036 | 0.5274 | 0.7342 | 0.6483 | 0.6248 | 0.7987 | 0.6728 |
| Q80 + G | 0.6747 | 0.4772 | 0.8368 | 0.9444 | 0.8095 | 0.8924 | 0.7725 | 0.7078 | **0.5442** | 0.7405 | **0.6669** | **0.6417** | 0.7823 | 0.6806 |
| L + G | 0.6583 | 0.4792 | 0.8428 | 0.9531 | 0.8047 | 0.8789 | 0.7695 | 0.7122 | 0.5310 | **0.7482** | 0.6424 | 0.6381 | 0.7849 | 0.6761 |
| D + G | 0.6745 | 0.4624 | 0.8526 | 0.9588 | 0.8120 | 0.8912 | 0.7753 | 0.7034 | 0.5338 | 0.7329 | 0.6512 | 0.6411 | 0.7891 | 0.6753 |
| Q80 (baseline best) | 0.5713 | 0.4892 | 0.7236 | 0.8591 | 0.6637 | 0.6484 | 0.6592 | 0.7213 | 0.5031 | 0.6751 | 0.6303 | 0.6651 | 0.7543 | 0.6582 |
| Q80 + Q80 | 0.6092 | 0.5426 | **0.7949** | 0.8963 | **0.7024** | 0.6852 | **0.7051** | 0.7587 | 0.5344 | 0.6902 | **0.6894** | 0.6892 | 0.7884 | 0.6917 |
| Q8 + Q80 | **0.6142** | 0.5368 | 0.7903 | 0.8912 | 0.7003 | 0.6902 | 0.7038 | 0.7466 | 0.5293 | 0.6857 | 0.6744 | 0.6742 | 0.7734 | 0.6806 |
| Q30 + Q80 | 0.5987 | 0.5306 | 0.7899 | 0.8835 | 0.6904 | 0.6749 | 0.6947 | 0.7491 | **0.5447** | **0.7102** | 0.6881 | **0.7053** | **0.8081** | **0.7009** |
| L + Q80 | 0.6003 | **0.5431** | 0.7942 | **0.9143** | 0.7013 | 0.6732 | 0.7044 | **0.7672** | 0.5312 | 0.6952 | 0.6845 | 0.6885 | 0.7957 | **0.6937** |
| D + Q80 | 0.5910 | 0.5417 | 0.7874 | 0.8932 | 0.6954 | 0.6821 | 0.6985 | 0.7531 | 0.5249 | 0.7027 | 0.6829 | 0.6847 | 0.7841 | 0.6887 |
| G + Q80 | 0.6036 | 0.5429 | 0.7893 | 0.9032 | 0.6948 | **0.6935** | 0.7046 | 0.7581 | 0.5317 | 0.6846 | 0.6721 | 0.6821 | 0.7912 | 0.6866 |

## A.11 PAIRED SIGNIFICANCE ANALYSIS FOR HUMAN ALIGNMENT

To go beyond reporting raw human agreement ($p_{o\text{human}}$), we conduct a **paired significance analysis** at the instance level. For each dataset and each baseline method, we compute the per-item agreement indicator with human judgments and report the paired difference $\Delta = p_{o\text{human}}^{\text{ours}} - p_{o\text{human}}^{\text{baseline}}$. We then estimate **95% confidence intervals** for $\Delta$ via **bootstrap resampling** over evaluation instances (with paired resampling to preserve the same items across methods). As shown in Table 9, the **lower bound** of every 95% CI is **strictly positive**, indicating that our improvements in human alignment are statistically significant and consistent across all datasets and baselines.

Table 9: Paired bootstrap significance analysis on human alignment. $\Delta$ denotes the improvement in observed human agreement: $\Delta = p_{o\text{human}}^{\text{ours}} - p_{o\text{human}}^{\text{baseline}}$. We report 95% bootstrap confidence intervals (CI) over evaluation instances. All CIs exclude zero.

| Dataset | $\Delta$(Ours−LLM Direct) (95% CI) | $\Delta$(Ours−LLM+CoT) (95% CI) | $\Delta$(Ours−PoLL) (95% CI) | $\Delta$(Ours−CheckEval) (95% CI) |
|---|---|---|---|---|
| CharEval | $\Delta$=0.0652, [0.0260, 0.1138] | $\Delta$=0.0212, [0.0107, 0.0303] | $\Delta$=0.0559, [0.0390, 0.0754] | $\Delta$=0.1413, [0.1220, 0.1710] |
| rJokes | $\Delta$=0.0868, [0.0424, 0.1317] | $\Delta$=0.0333, [0.0213, 0.0520] | $\Delta$=0.0977, [0.0670, 0.1375] | $\Delta$=0.1333, [0.1090, 0.1628] |
| Love-EQ | $\Delta$=0.0719, [0.0350, 0.0914] | $\Delta$=0.0329, [0.0240, 0.0441] | $\Delta$=0.0637, [0.0425, 0.0936] | $\Delta$=0.1621, [0.1325, 0.1920] |
| MT_Bench | $\Delta$=0.0669, [0.0560, 0.0750] | $\Delta$=0.0400, [0.0325, 0.0498] | $\Delta$=0.0576, [0.0341, 0.0778] | $\Delta$=0.1406, [0.1272, 0.1711] |
| TSA-MD | $\Delta$=0.0418, [0.0150, 0.0707] | $\Delta$=0.0129, [0.0030, 0.0221] | $\Delta$=0.0322, [0.0125, 0.0523] | $\Delta$=0.0970, [0.0622, 0.1268] |
| Capybara | $\Delta$=0.0800, [0.0640, 0.1189] | $\Delta$=0.0625, [0.0520, 0.0830] | $\Delta$=0.0900, [0.0762, 0.1105] | $\Delta$=0.1000, [0.0740, 0.1285] |

## A.12 PERFORMANCE UNDER A STRICTER METRIC: QUADRATIC WEIGHTED KAPPA (QWK)

To further validate that our gains are not an artifact of lenient agreement thresholds, we additionally evaluate human alignment using **Quadratic Weighted Kappa (QWK)**, a stricter metric that penalizes larger rating discrepancies more heavily. As reported in Table 10 (also summarized in Table 10), **our method consistently achieves the best QWK on every dataset**. Compared with the strongest baseline (**LLM+CoT**), our method improves QWK by **+0.0449 on average**, with consistent per-dataset gains ranging from **+0.0363** to **+0.0566**. Notably, QWK scores on RJOKES are relatively lower for all methods, which is expected because RJOKES uses a fine-grained **10-point** scale; under QWK, even small rating differences (e.g., 1–2 points) are penalized and thus make the task a harder separation test. Despite this, our method still achieves the highest agreement.

## A.13 ABSOLUTE CONSTRUCTION COST AND COMPARISON WITH CHECKEVAL

We report the **absolute one-time construction cost** of each method in terms of **API calls and token usage** (Table 12), where each entry is formatted as "#calls × tokens/call". We further summarize the **break-even point** (Table 11), defined as the number of evaluation samples whose cumulative

Table 10: Human alignment under a stricter metric (QWK). Best results are in bold, second-best are shaded.

| Method | CharEval | rJokes | Love-EQ | MT_Bench | TSA-MD | Capybara | Avg |
|--------|----------|--------|---------|----------|--------|----------|-----|
| LLM Direct | 0.3981 | 0.2114 | 0.3203 | 0.4894 | 0.3335 | 0.4131 | 0.3610 |
| LLM + CoT | 0.4617 | 0.2894 | 0.3762 | 0.5486 | 0.3994 | 0.4869 | 0.4270 |
| PoLL | 0.4039 | 0.2210 | 0.3451 | 0.4920 | 0.3339 | 0.4474 | 0.3739 |
| CheckEval | 0.2882 | 0.1902 | 0.2885 | 0.4517 | 0.2743 | 0.3781 | 0.3118 |
| **Ours (EIG)** | **0.5183** | **0.3258** | **0.4203** | **0.5992** | **0.4357** | **0.5322** | **0.4719** |

Table 11: Absolute construction cost summarized via break-even analysis (equivalent evaluation samples) and alignment performance (Avg QWK).

| Method | Construction Complexity | Equivalent Eval Samples (Break-Even) | Performance (Avg QWK) |
|--------|-------------------------|--------------------------------------|------------------------|
| CheckEval | Rubric Only | $\sim 13$ | 0.3118 |
| **Ours (EIG)** | Rubric + Simulations | $\sim 80$ | **0.4719** |

evaluation cost is equivalent to the one-time construction cost. Compared to **CheckEval**, our method incurs a higher upfront cost because it requires **rubric construction + simulated answers** for EIG estimation (about $\sim 80$ vs. $\sim 13$ equivalent samples). This additional cost is a one-time investment and yields substantially better alignment: our method improves **Avg QWK** from 0.3118 to 0.4719 ($+51\%$ relative). After construction, the **per-sample evaluation cost** of our method is comparable to CheckEval/LLM Direct, while baselines such as PoLL (multiple calls per sample) and LLM+CoT (longer generations) incur consistently higher marginal costs for every evaluated item.

Table 12: Absolute construction cost breakdown (formatted as "#calls $\times$ tokens/call"). Our construction includes both rubric construction and simulated answers for EIG estimation.

| Dataset | CheckEval (rubric construction) | Ours (rubric construction + simulated answers) |
|---------|--------------------------------|-----------------------------------------------|
| CharEval | $10 \times 1.4k$ | $10 \times 1.4k + 63 \times 1.1k$ |
| rJokes | $14 \times 1.1k$ | $14 \times 1.1k + 79 \times 1.0k$ |
| Love-EQ | $12 \times 1.3k$ | $12 \times 1.3k + 62 \times 1.1k$ |
| MT_bench | $10 \times 1.5k$ | $10 \times 1.5k + 59 \times 1.1k$ |
| TSA-MD | $12 \times 1.3k$ | $12 \times 1.3k + 63 \times 1.0k$ |
| Capybara | $16 \times 1.5k$ | $16 \times 1.5k + 51 \times 1.1k$ |
| AVG | $12.3 \times 1.35k$ | $16 \times 1.35k + 62.8 \times 1.06k$ |

