# OpenReview forum: "Information-Theoretic Questionnaire Construction for Consistent Evaluation of Subjective Tasks with LLMs"
_ICLR.cc/2026/Conference — ICLR 2026 Conference Desk Rejected Submission_

### Official Review · Reviewer_5TSt · 2025-10-16

**Soundness:** 3
**Presentation:** 3
**Contribution:** 3
**Rating:** 8
**Confidence:** 3

**Summary:**

The paper tackles the instability (dimension drift) of free-form evaluation in LLM-as-a-judge by generating auxiliary evaluation questions. The questions are selected by maximizing the expected information gain. In experiments, the method does not only improve self-consistency, but also enhancing human alignment.

**Strengths:**

The paper formulates subjective evaluation into a quantitative information-theoretic problem, which is the main originality of the paper. The paper clearly presents the details of their approach, with codebase for reproducibility. The paper resolves the instability in evaluation, contributing to the reliability of LLM-as-a-judge.

**Weaknesses:**

The proposed method fails to provide theoretical explanations on why it improves alignment with human evaluation. The method introduces additional hyperparameters to tune, e.g., simulation round $m$, as well as the size of candidate question set $Q$.

**Questions:**

1. The simulated response filling procedure (Eq. (4)) is confusing. $p(a\mid q, I)$ is instance-dependent, but the approximation uses $m$ different instances. How are the instances sampled?
2. In Algorithm 1, Line 1, $Q$ is generated from $D$. Then are all instances in $D$ attached in the prompt template of Figure 9. If so, will the prompt be too long to fit in the context length of LLM?
3. How is $m$ and size of $Q$ determined/chosen in practice?
4. While the improvement of self-consistency is reasonable, as the optimization objective is to reduce uncertainty in the rating, can you explain why it also improves alignment with human evaluation. The reason is unclear as human annotators do not follow the rubrics generated via EIG framework.
5. How does the proposed method compare with LLM Direct + majority voting, especially in human judgement alignment? In other words, we run repeated LLM-as-a-judge evaluation on each instance, and choose the most-frequent rating as the final rating. This trick can be applied on top of any existing evaluation approach, and can clearly improve self-consistency with low cost.
6. Typo in prompt template (Figure 9-11): Instrution -> Instruction.

---

> ### Author Response · Authors · 2025-11-26
> **Response to Reviewer 5TSt (Part 1/2)**
>
> ## Weaknesses
> ---
> ### W1. Lack of theoretical grounding
>
> We appreciate the reviewer for pointing out the need for deeper theoretical grounding. Our method's effectiveness is not accidental but rooted in established **cognitive science theories regarding human inquiry**.
>
> **1. Theoretical Basis: Optimal Data Selection**
> A substantial body of work in cognitive psychology suggests that human evidence-seeking is driven by **maximizing the expected reduction of uncertainty**.
>
> - **Oaksford & Chater (1994)** formalized the classic Wason selection task as an **Optimal Data Selection** problem, demonstrating that humans intuitively select evidence (data points) that maximizes **Expected Information Gain (EIG)** [1].
> - **Rothe et al. (2017)** further extended this to **question-asking**, showing that EIG is a primary predictor of which questions humans naturally choose to ask to distinguish between hypotheses [2].
>
> **2. Why This Improves Alignment**
> Our framework mathematically formalizes this cognitive principle. By selecting evaluation questions with the highest EIG, we are essentially mimicking the **human cognitive strategy** of identifying the most "discriminative" features. This alignment in the *process* of inquiry (seeking the most informative evidence) leads to the observed alignment in the *result* of evaluation.
>
> **References**
>
> - \[1\] Oaksford, M., & Chater, N. *A rational analysis of the selection task as optimal data selection*. **Psychological Review**, 101(4), 608, 1994.
> - \[2\] Rothe, A., Lake, B. M., & Gureckis, T. *Question asking as program generation*. **NeurIPS**, 30, 2017.
>
> ---
>
> ### W2. Extra hyperparameters (number of simulations \(m\), candidate pool size \(|Q|\))
>
> In practice, neither the number of simulations \(m\) nor the candidate pool size \(|Q|\) requires careful manual tuning. Instead, we determine both automatically via a simple **“stop when converged”** principle:
>
> - **Choosing \(|Q|\)**: We gradually expand the question pool and monitor its growth until it **converges** according to a rule-based criterion; once convergence is detected, we stop, which implicitly determines \(|Q|\).
> - **Choosing \(m\)**: We run **online simulation and EIG computation**, continuously tracking each question’s EIG values and the induced ranking. When the EIG scores and rankings become **stable**, we stop the simulations, which indirectly determines the required \(m\).
>
> Figure 4 provides convergence curves for the candidate set and monitoring signals for EIG-ranking stabilization on real tasks.
>
> ---
>
> ## Questions
> ---
> ### Q1. When estimating $p(a \mid q, I)$using \(m\) different instances, how are these instances sampled?
>
> In the main experiments, we construct simulated contexts by **uniformly sampling instances from the dataset \(D\)** at random, as an approximation to the overall distribution.
>
> We also add an ablation study **(see Appendix A.8)** comparing random sampling against a **“cluster-then-stratified sampling”** strategy. The results show that the resulting **EIG rankings are highly consistent** under both schemes, so we adopt the simpler random sampling in the main pipeline.
>
> ---
>
> ### Q2. In Algorithm 1, \(Q\) is generated from \(D\). Does this mean you must include all instances of \(D\) in the prompt? Would that exceed the LLM context length?
>
> In the **question generation stage**, we use a **batched few-shot** procedure:
>
> - (a) Each prompt includes only a **small mini-batch** of examples to generate questions relevant to that batch.
> - (b) After each batch, we perform **residual-style deduplication and merging**, and iterate until we obtain the full candidate pool \(Q\).
>
> Therefore, we never place the entire dataset \(D\) into a single prompt, and the approach does not run into context-length issues.

---

> ### Author Response · Authors · 2025-11-26
> **Response to Reviewer 5TSt (Part 2/2)**
>
> ### Q3. How do you choose \(m\) and \(|Q|\) in practice?
>
> Same as in **W2**: we determine both automatically using the **“stop when converged”** criterion, rather than fixing them as manually tuned constants.
>
> ---
>
> ### Q4. Why would reducing self-consistency uncertainty improve agreement with human evaluation, given that humans do not strictly follow the EIG-derived rubric?
>
> The improvement in human alignment, despite humans not explicitly using our rubric, is driven by two key factors: **theoretical feature alignment** and **empirical universality**.
>
> **1. Theoretical Alignment: Capturing Discriminative Features**
> While humans do not consult the generated rubric, the EIG framework mathematically identifies the **latent dimensions** that are most effective at distinguishing response quality. These dimensions (e.g., logical coherence, instruction following) act as proxies for the internal criteria humans implicitly use. By reducing uncertainty on these "deciding factors," we force the model to align with the underlying quality signals that matter to humans.
>
> **2. Empirical Evidence: Universality of Dimensions (Heterogeneous Experiments)**
> Our **heterogeneous experiments** (see Response to Reviewer bGb2 and  **Table 8**) provide strong empirical support for this claim. We observed that rubrics generated by one model (e.g., Model A) significantly improve the human alignment of a different model (e.g., Model B).
>
> - **Implication:** This cross-model transferability proves that the EIG-selected questions are not merely capturing model-specific biases or artifacts. Instead, they capture **objective, universal quality dimensions** that are recognized by different diverse models.
>
> ---
>
> ### Q5. How does your method compare to “direct majority voting” in LLM-as-a-judge?
>
> We have rigorously evaluated this exact strategy. Our baseline **PoLL (same)** is specifically implemented to match the reviewer’s suggestion: it aggregates multiple sampling runs from the same model via majority voting. Additionally, we report **PoLL (different)**, which aggregates votes across diverse model families. As detailed in the table below, the results reveal a critical limitation: while majority voting improves self-consistency, it yields **negligible gains in human alignment** (e.g., **PoLL (same)** shows minimal improvement over **LLM Direct**). In contrast, our **EIG method** consistently outperforms these baselines. This disparity stems from the fact that Majority Voting effectively mitigates *random noise* but fails to address *systematic biases*; when models exhibit consistent errors—such as misunderstanding nuances or favoring specific styles—voting simply converges to the incorrect answer with higher confidence. Conversely, our EIG method actively **calibrates the reasoning process** by leveraging high-information questions (discriminative criteria) to force the model to focus on the specific dimensions that drive human judgments, thereby correcting these biases and achieving significant alignment gains.
>
> **Table 7:** Comparison of self-consistency $\kappa_{\text{self}}$, measured by Cohen's $\kappa$ and human alignment ($p_{o\text{human}}$, measured by observed agreement). Best results are in bold, and second-best are underlined.
> | Method | $\kappa_{\text{self}}$ | | | | | | | $p_{o\text{human}}$ | | | | | | |
> | --- | ---:| ---:| ---:| ---:| ---:| ---:| ---:| ---:| ---:| ---:| ---:| ---:| ---:| ---:|
> | | CharEval | rJokes | Love\_EQ | MT\_bench | TSA\_MD | Capybara | Avg | CharEval | rJokes | Love\_EQ | MT\_bench | TSA\_MD | Capybara | Avg |
> | LLM Direct | 0.6218 | 0.4420 | 0.7924 | 0.9314 | 0.7768 | 0.8519 | 0.7361 | 0.6534 | 0.4526 | 0.6736 | 0.5864 | 0.5949 | 0.7125 | 0.6122 |
> | PoLL(different) | $\underline{0.6317}$ | $\underline{0.4479}$ | 0.7834 | $\underline{0.9407}$ | 0.7714 | 0.8497 | 0.7375 | $\underline{0.6627}$ | 0.4417 | $\underline{0.6818}$ | $\underline{0.5957}$ | $\underline{0.6045}$ | 0.7025 | $\underline{0.6148}$ |
> | PoLL(same) | 0.6264 | 0.4457 | $\underline{0.7947}$ | 0.9346 | $\underline{0.7790}$ | $\underline{0.8528}$ | $\underline{0.7389}$ | 0.6541 | $\underline{0.4532}$ | 0.6783 | 0.5869 | 0.5977 | $\underline{0.7142}$ | 0.6141 |
> | **Ours(EIG)** | **0.6728** | **0.4832** | **0.8513** | **0.9523** | **0.8145** | **0.8974** | **0.7786** | **0.7186** | **0.5394** | **0.7455** | **0.6533** | **0.6367** | **0.7925** | **0.6810** |
>
> ---
>
> ### Q6. Spelling error in the prompt template (Instrution → Instruction)
>
> Thanks for catching this. We have corrected the typo in the revised manuscript.

---

### Official Review · Reviewer_egqB · 2025-10-26

**Soundness:** 3
**Presentation:** 3
**Contribution:** 3
**Rating:** 4
**Confidence:** 3

**Summary:**

The paper addresses the reproducibility crisis in subjective LLM evaluation by identifying and formalizing the root cause as dimension drift. It proposes an Expected Information Gain (EIG)-based framework that constructs a stable, adaptive rubric via a two-stage “generate–then–score” process: first generating diverse candidate evaluation questions, then selecting the most informative subset using EIG. Evaluated on six subjective benchmarks (e.g., CharacterEval, The rJokes, Love-EQ), the method significantly improves both self-consistency (Cohen’s κ) and alignment with human judgments over CoT and fixed-rubric baselines.

**Strengths:**

1. Dimension drift is precisely defined and empirically validated as a core instability mechanism in LLM-as-a-judge systems.

2. Reformulating rubric construction as a Bayesian experimental design (BED) problem with EIG provides a principled information-theoretic foundation.

3. The proposed “stability–informativeness loop” resolves the trade-off between rigid checklists and unstable free-form reasoning.

4. Introduces EMdim to quantify drift and amortized cost analysis, offering actionable insights for real-world deployment.

**Weaknesses:**

1.  The joint distribution p^(y,a∣q,I) is estimated entirely via LLM simulations, which may inherit or amplify the very drift the method aims to eliminate.

2. As a newly introduced dataset, Love-EQ lacks description of annotation protocol, inter-annotator agreement, or scale, weakening reproducibility.

3. Using uncorrected observed agreement (rather than weighted κ or correlation) may overstate alignment, especially with lenient tolerance thresholds (±1 on 5-point scale).

4. While amortized cost is analyzed, the absolute upfront cost (e.g., LLM calls for rubric construction) is not quantified or compared to alternatives like CheckEval.

**Questions:**

1. How sensitive is EIG ranking to the choice of base LLM used for simulation? Could a drifted LLM produce misleading EIG scores?

2. Can the authors provide more details about Love-EQ (e.g., number of samples, annotator count, Fleiss'k)?

3. Why was observed agreement chosen over standard metrics like Spearman correlation or weighted Cohen’s κ for human alignment?

4. Is the rubric construction process reusable across domains, or does it require full re-generation for each new task?

---

> ### Author Response · Authors · 2025-11-26
> **Response to Reviewer egqB (Part 1/4)**
>
> ## Weaknesses
> ---
> ### W1. The joint distribution simulated by LLMs may inherit or amplify model bias/drift
>
> ## **1. Theoretical Robustness: EIG Measures Statistical Dependence**
>
> Theoretically, EIG (IG(q,a)=H(Y)−H(Y∣q,a)) measures the **Mutual Information** between the answer and the score, focusing on statistical correlation rather than absolute probabilities. This formulation naturally mitigates drift:
>
> - **Noise is Penalized:** If drift manifests as random fluctuations uncorrelated with quality, it increases conditional entropy H(Y∣q,a), thereby **decreasing IG**. Random noise is thus naturally filtered out rather than amplified.
> - **Systematic Bias is Offset:** Systematic drifts (e.g., score inflation) typically shift both H(Y) and H(Y∣q,a) simultaneously. These effects tend to **offset each other** during subtraction, preserving the relative ranking of questions.
> - **Structure Extraction:** As long as the true **statistical dependency** (e.g., "hallucination → score drop") remains, EIG stably extracts the task's **discriminative structure** rather than inheriting the model's raw output tendencies.
>
> ###### **2. Empirical Validation: Cross-Model Robustness**
>
> "To empirically verify that our method does not overfit to specific simulator artifacts, we conducted **heterogeneous evaluations** (detailed in **Appendix A.10 (Table 8)** of the revised paper), where the EIG estimator and the final evaluator are entirely different models (e.g., Llama-3.3 for EIG estimation, GPT-4.1 for evaluation). (We also discuss this analysis in our response to Reviewer bGb2)."
>
> - **Results:** The performance gap between homogeneous (same-model) and heterogeneous (cross-model) settings is negligible.
> - **Implication:** This confirms that the evaluation dimensions extracted by EIG are **universal and transferable**, effectively ruling out the hypothesis that the method inherits model-specific biases.
>
> #### **3. Empirical Stability: Rapid Convergence of EIG Estimation**
>
> **Figure 4(b)** provides direct evidence that our EIG estimation is robust against simulator drift.
>
> - **Convergence:** As the number of simulated responses increases, the normalized EIG values for different questions quickly **converge to stable trajectories**.
> - **Ranking Stability:** The relative ranking of questions stabilizes with growing sample size, exhibiting substantially reduced fluctuation compared to the early sampling stages.
> - **Noise Reduction:** The magnitude of stochastic variation **diminishes noticeably** as more samples are accumulated.
>
> ---
>
> ### W2. Love-EQ lacks annotation details (protocol, inter-annotator agreement, scale), which harms reproducibility
>
> Thanks for pointing this out. In the revised manuscript, we will expand the Appendix to include: (a) the annotation workflow and detailed protocol **(see Appendix A.7)**, (b) the dataset size and number of annotators **(1,039 instances annotated by three annotators)**, and (c) the inter-annotator agreement, measured by **Fleiss’ κ (κ = 0.7326)** computed over the three annotators’ ratings.
>
> In addition, we will publicly release the complete Love-EQ dataset and the full annotation guideline document in our open-source GitHub repository to facilitate reproducibility and enable follow-up research.

---

> ### Author Response · Authors · 2025-11-26
> **Response to Reviewer egqB (Part 2/4)**
>
> ### W3. Using *uncorrected observed agreement* may overestimate alignment
>
> We appreciate the reviewer’s rigorous suggestion. We agree that simple observed agreement with a tolerance threshold might mask subtle discrepancies. To address this, we have augmented our evaluation with **Quadratic Weighted Cohen’s κ (QWK)**, a stricter metric that penalizes disagreements based on their magnitude and **does not use any tolerance threshold**.
>
> #### **1. Performance under Stricter Metric (QWK)**
>
> As shown in the table below (and **Table 10** in the revision), our method consistently outperforms all baselines under the QWK metric across all datasets.
>
> - **Robust Superiority:** Our method surpasses the strongest baseline (LLM+CoT) by an average of **+0.0449**, with consistent gains on every single dataset (ranging from +0.0363 to +0.0566).
> - **Independence from Tolerance:** These results confirm that our improvements are genuine and robust, not artifacts of a "lenient threshold."
> - **Note on rJokes:** The relatively lower QWK scores on *rJokes* across all methods are expected. This task uses a fine-grained **10-point scale**. Without tolerance thresholds, small discrepancies (e.g., 1-2 points) significantly impact the metric, making it a harder separation test. Even in this challenging setting, our method achieves the highest agreement.
>
> **Table 10:** Human alignment under a stricter metric (QWK). Best results are in bold, and second-best are underlined.
> | Method | CharEval | rJokes | Love-EQ | MT\_Bench | TSA-MD | Capybara | Avg |
> | --- | ---: | ---: | ---: | ---: | ---: | ---: | ---: |
> | LLM Direct | 0.3981 | 0.2114 | 0.3203 | 0.4894 | 0.3335 | 0.4131 | 0.3610 |
> | LLM + CoT | $\underline{0.4617}$ | $\underline{0.2894}$ | $\underline{0.3762}$ | $\underline{0.5486}$ | $\underline{0.3994}$ | $\underline{0.4869}$ | $\underline{0.4270}$ |
> | PoLL | 0.4039 | 0.2210 | 0.3451 | 0.4920 | 0.3339 | 0.4474 | 0.3739 |
> | CheckEval | 0.2882 | 0.1902 | 0.2885 | 0.4517 | 0.2743 | 0.3781 | 0.3118 |
> | **Ours(EIG)** | **0.5183** | **0.3258** | **0.4203** | **0.5992** | **0.4357** | **0.5322** | **0.4719** |
>
> #### **2. Statistical Significance Analysis**
>
> To further ensure these gains are not due to chance, we conducted paired bootstrap significance tests (1000 resamples). The ** new** table 10 below reports the improvements (Δ) of our method over baselines along with **95% Confidence Intervals (CI)**. The **lower bounds of all 95% confidence intervals are strictly above 0**, indicating that our performance gains are **statistically significant** across all datasets and against all baselines.
>
> **Table 9:** Paired bootstrap significance analysis on human alignment across datasets.
> | Dataset | Δ(Ours−LLMDirect)(95%CI) | Δ(Ours−LLM+CoT)(95%CI) | Δ(Ours−PoLL)(95%CI) | Δ(Ours−CheckEval)(95%CI) |
> | --- | --- | --- | --- | --- |
> | CharEval | Δ=0.0652,95%CI[0.0260,0.1138] | Δ=0.0212,95%CI[0.0107,0.0303] | Δ=0.0559,95%CI[0.0390,0.0754] | Δ=0.1413,95%CI[0.1220,0.1710] |
> | rJokes | Δ=0.0868,95%CI[0.0424,0.1317] | Δ=0.0333,95%CI[0.0213,0.0520] | Δ=0.0977,95%CI[0.0670,0.1375] | Δ=0.1333,95%CI[0.1090,0.1628] |
> | Love-EQ | Δ=0.0719,95%CI[0.0350,0.0914] | Δ=0.0329,95%CI[0.0240,0.0441] | Δ=0.0637,95%CI[0.0425,0.0936] | Δ=0.1621,95%CI[0.1325,0.1920] |
> | MT\_Bench | Δ=0.0669,95%CI[0.0560,0.0750] | Δ=0.0400,95%CI[0.0325,0.0498] | Δ=0.0576,95%CI[0.0341,0.0778] | Δ=0.1406,95%CI[0.1272,0.1711] |
> | TSA-MD | Δ=0.0418,95%CI[0.0150,0.0707] | Δ=0.0129,95%CI[0.0030,0.0221] | Δ=0.0322,95%CI[0.0125,0.0523] | Δ=0.0970,95%CI[0.0622,0.1268] |
> | Capybara | Δ=0.0800,95%CI[0.0640,0.1189] | Δ=0.0625,95%CI[0.0520,0.0830] | Δ=0.0900,95%CI[0.0762,0.1105] | Δ=0.1000,95%CI[0.0740,0.1285] |

---

> ### Author Response · Authors · 2025-11-26
> **Response to Reviewer egqB (Part 3/4)**
>
> ### W4. Only amortized cost is analyzed; the absolute cost of rubric construction is not quantified, nor compared with CheckEval
>
> We have added a detailed breakdown of the **absolute construction costs** in **Table 11**, reporting both API calls and token usage. Each entry is formatted as `"calls × tokens/call"`.
>
> #### **1. Absolute Cost & Comparison with CheckEval**
>
> As shown in the table below, we quantify the "Break-Even Point" — the number of evaluation samples equivalent to the one-time construction cost.
>
> **Table 11:** Absolute construction cost summarized via break-even analysis (equivalent evaluation samples) and alignment performance (Avg QWK).
> | Method | Construction Complexity | Equivalent Eval Samples(Break-Even) | Performance(Avg QWK) |
> | --- | --- | --- | --- |
> | **CheckEval** | Rubric Only | ~13 samples | 0.3118 |
> | **Ours(EIG)** | Rubric+Simulations | ~80 samples | **0.4719**(+51%) |
>
> - **Cost Difference:** Our construction cost is indeed higher than CheckEval (~80 vs. ~13 equivalent samples) due to the simulation step necessary for EIG calculation.
> - **Value Proposition:** This additional upfront cost is a **one-time investment**. In exchange for an equivalent of ~67 extra evaluation samples, we achieve a **+51% relative improvement** in alignment (QWK 0.4719 vs. 0.3118) compared to CheckEval. For any real-world benchmark containing hundreds or thousands of examples, this trade-off is highly favorable.
>
> #### **2. Long-term Efficiency (Amortization)**
>
> Once the rubric is constructed, the **per-sample evaluation cost** of our method is comparable to CheckEval and LLM Direct.
>
> - In contrast, baselines like **PoLL** (multiple calls per sample) and **LLM+CoT** (long output generation) incur **continuously higher marginal costs** for every new sample evaluated.
> - Therefore, for any dataset larger than ~80 samples, our method rapidly becomes more cost-effective than PoLL/CoT while maintaining superior accuracy.
>
> **Table 12:** Absolute construction cost breakdown (formatted as ``calls $\times$ tokens/call''). Our construction includes both rubric construction and simulated answers for EIG estimation.
> | Dataset | CheckEval(rubricconstruction) | Ours(rubricconstruction + simulatedanswers) |
> | --- | --- | --- |
> | CharEval | 10×1.4k | 10×1.4k + 63×1.1k |
> | rJokes | 14×1.1k | 14×1.1k + 79×1.0k |
> | Love-EQ | 12×1.3k | 12×1.3k + 62×1.1k |
> | MT\_bench | 10×1.5k | 10×1.5k + 59×1.1k |
> | TSA-MD | 12×1.3k | 12×1.3k + 63×1.0k |
> | Capybara | 16×1.5k | 16×1.5k + 51×1.1k |
> | AVG | 12.3×1.35k | 16×1.35k + 62.8×1.06k |

---

> ### Author Response · Authors · 2025-11-26
> **Response to Reviewer egqB (Part 4/4)**
>
> ## Questions
> ---
> ### Q1. How sensitive is EIG ranking to the baseline LLM used for simulation? Can model drift cause misleading EIG?
>
> EIG ranking is **highly insensitive** to the choice of base LLM, and a "drifted" model does **not** produce misleading scores.
>
> As detailed in our **Response to Reviewer bGb2, we have specifically addressed this by conducting extensive  heterogeneous experiments** where the model used for simulation (EIG estimation) differs from the evaluator.
>
> Our findings (referring to **Table 8**) demonstrate two key points regarding sensitivity and drift:
>
> 1. **Insignificant Sensitivity:** The performance gap when swapping the simulator is negligible. For instance, using **Qwen3-Next-80B** instead of **GPT-4.1** to simulate answers for a GPT-4.1 evaluator results in a Human Alignment (ρhuman​) difference of **less than 0.001**. This proves that the EIG rankings remain stable even when the simulator changes significantly.
> 2. **Robustness to Drift:** Far from being misleading, a "drifted" (different) simulator can sometimes yield even *better* evaluation criteria (e.g., Llama-3.3 guiding Qwen3-Next-80B outperforms Qwen3-Next-80B guiding itself).
>
> This confirms that EIG captures **universal, task-intrinsic properties** rather than model-specific artifacts. Please see Response to Reviewer bGb2 for the complete experimental data and analysis.
>
> ---
>
> ### Q2. Can you provide more details about Love-EQ (sample size, number of annotators, Fleiss’ κ, etc.)?
>
> Thanks for pointing this out. In the revised manuscript, we will expand the Appendix to include: (a) the annotation workflow and detailed protocol **(see Appendix A.7)**, (b) the dataset size and number of annotators **(1,039 instances annotated by three annotators)**, and (c) the inter-annotator agreement, measured by **Fleiss’ κ (κ = 0.7326)** computed over the three annotators’ ratings.
>
> In addition, we will publicly release the complete Love-EQ dataset and the full annotation guideline document in our open-source GitHub repository to facilitate reproducibility and enable follow-up research.
>
> ---
>
> ### Q3. Why use observed agreement instead of Spearman correlation or weighted Cohen’s κ to measure human alignment?
>
> We originally reported observed agreement (with tolerance) to provide an intuitive measure of **absolute score precision** (i.e., measuring exact or adjacent matches), which complements ranking-based correlations. However, we agree that QWK is strictly more robust against chance. As detailed in **W3**, we have added **Quadratic Weighted Cohen’s κ (QWK)** results. Our method consistently outperforms baselines under this stricter metric.  Furthermore, we performed paired-bootstrap significance tests. The **95% Confidence Intervals (CIs)** for the improvements (Δ) in **both metrics** consistently exclude zero, confirming that the gains are **statistically significant**.
>
> ---
>
> ### Q4. Can the rubric construction process be reused across domains?
>
> Our experiments indicate that the rubric construction pipeline is **reusable across domains**, rather than requiring re-design from scratch for each new task. Within the same framework, we systematically validate on **six subjective evaluation datasets** spanning diverse domains: **CharacterEval** (Chinese role-play dialogue), **rJokes** (English humor scoring), **Love-EQ** (emotion intelligence in romantic-chat scenarios; newly introduced in this work), **MT-Bench** (general multi-turn QA quality), **TSA-MD** (multi-domain sentiment and emotion intensity), and **Capybara-pref1** (broad instruction-following and dialogue preference). Across all these domains, our method consistently outperforms existing SOTA, demonstrating that the same rubric construction and alignment procedure can be **directly transferred** to new subjective evaluation tasks by only replacing task descriptions and dimension definitions, without redesigning the entire pipeline.

---

### Official Review · Reviewer_bGb2 · 2025-10-29

**Soundness:** 3
**Presentation:** 3
**Contribution:** 2
**Rating:** 2
**Confidence:** 3

**Summary:**

The paper addresses the reproducibility and reliability issue in LLM-as-a-Judge evaluation, identifying dimension drift—a phenomenon where a model’s implicit evaluation criteria shift across repeated judgments—as a major cause of inconsistency. To mitigate this, the authors propose an information-theoretic framework that reformulates subjective evaluation as an Expected Information Gain (EIG) maximization problem. Specifically, the method first generates diverse candidate evaluation questions (representing potential judgment dimensions) and then selects the top-k most informative ones to construct a compact and fixed questionnaire. This structured questionnaire replaces free-form reasoning in evaluation, ensuring consistent judgment dimensions.
Experiments on six benchmarks—CharacterEval, rJokes, Love-EQ, MT-Bench, TSA-MD, and Capybara-Pref—demonstrate improved self-consistency (Cohen’s κ, +7.6% over CoT), better human alignment, and lower amortized cost compared to Direct, CoT, CheckEval, and PoLL baselines. The authors further introduce EMdim, a metric to quantify dimensional consistency, and show that binary question formats yield higher stability.

**Strengths:**

1.The paper tackles an important and often overlooked issue in LLM-as-a-Judge, that models can change their implicit judging criteria across runs. Framing this as “dimension drift” is both intuitive and empirically grounded, and it helps explain inconsistencies that prior work only described superficially.
2. Turning subjective evaluation into an information-gain maximization problem is an elegant move. The EIG-based questionnaire offers a structured and interpretable way to stabilize evaluations without over-engineering. It is easy to implement and could be reused across different subjective tasks.
3. Experiments are broad and reasonably convincing.The authors evaluate on six distinct benchmarks with multiple model families, and the analysis covers not only accuracy but also stability, cost, and dimensional consistency.

**Weaknesses:**

1. I noticed that the title in the PDF differs from the one shown on OpenReview. You might want to double-check that they are consistent.
2. The joint distributions $p\(y, a \| q, I\)$ and $p\( a \| q, I\)$ used for EIG estimation are simulated using the same or closely related LLMs that later act as evaluators. This creates a closed-loop bias, as the model essentially optimizes for its own preferences.
3.  This paper measures human alignment via raw agreement rates without significance testing or chance correction. This weakens claims of improvement.
4.The top-k question selection relies on greedy ranking without theoretical justification (e.g., submodularity or redundancy control).

**Questions:**

1.Are the same or closely related LLMs used to both estimate the EIG distributions and perform evaluation? If so, how might this closed-loop setup bias the questionnaire toward the model’s own preferences? Some results using heterogeneous models would clarify robustness.
2. Binary questions appear most stable, but possibly less sensitive to nuanced differences. Could you report quantitative trade-offs between stability and discriminative precision (e.g., AUROC or fine-grained agreement)?

---

> ### Author Response · Authors · 2025-11-26
> **Response to Reviewer bGb2 (Part 1/5)**
>
> ## Weaknesses
> ---
> ### W1. Inconsistent titles between PDF and OpenReview
>
> Thank you very much for noticing this inconsistency. The title in the OpenReview is the correct version, and we have revised the title on PDF accordingly.
>
> ---
>
> ### W2. Potential closed-loop bias from using the same / similar LLMs for EIG estimation and evaluation
>
> We thank the reviewer for raising this critical point. We agree that distinguishing between **model-specific bias** (self-preference) and **task-intrinsic criteria** (valid evaluation dimensions) is essential for the validity of our method. To prove that our EIG-based framework extracts universal evaluation dimensions rather than simply "overfitting" to the generator's preferences, we conducted a comprehensive **Cross-Model Validation** (see new **Table 8**).
>
> **1. Experimental Setup:  heterogeneous-pipeline**
> We decoupled the pipeline into two roles: the *Question Generator* (EIG Estimator) and the *Evaluator (questionnaire-evaluation )*. We permuted these roles across **6 distinct models** spanning varying capabilities and architectures:
>
> - **Proprietary:** GPT-4.1 (`G`);
> - **Open-Weights:** Llama-3.3-70B (`L`), Qwen3-8B/30B/Next-80B (`Q8`, `Q30`, `Q80`);
> - **Reasoning-Specialized:** DeepSeek-R1 (`D`).
>
> **2.Key Results: Universality over Bias**
>
> Our results strongly negate the closed-loop bias hypothesis. Across all heterogeneous configurations, we observe that both **Human Alignment (ρhuman​)** and **Self-Consistency (κself​)** remain highly stable compared to the homogeneous baselines.
> **1) Robust Transferability:** Taking the GPT-4.1 (`G`) evaluator as an example, the performance gap between the homogeneous setup (`G`+`G`) and the heterogeneous setup (`Q80`+`G`) is negligible across both key metrics:
>
> - **Human Alignment (ρhuman​ Avg):** **0.6810** (`G`+`G`) vs. **0.6806** (`Q80`+`G`) — *Gap < 0.001*.
> - **Self-Consistency (κself​ Avg):** **0.7786** (`G`+`G`) vs. **0.7725** (`Q80`+`G`) — *Gap < 0.01*.
>
> This consistent stability indicates that the "judging criteria" extracted by EIG are universally recognized and not sensitive to the generator's specific architecture.
>
> **2)Heterogeneous Can Outperform Homogeneous:**
>
> In the Qwen3-Next-80B (`Q80`) evaluation setting, using **Llama-3.3** (`L`) to generate questions (`L`+`Q80`) even yields higher human alignment scores (**0.6937**) than the homogeneous `Q80`+`Q80` baseline (**0.6917**), further confirming that our method captures task-intrinsic dimensions rather than model-specific biases.

---

> ### Author Response · Authors · 2025-11-26
> **Response to Reviewer bGb2 (Part 2/5)**
>
> **Table 8:** Heterogeneous-variant experiments using different LLMs in the two stages of our framework—EIG estimation and questionnaire-based evaluation. The table contains both G-based configurations and Q80-based configurations. Best results are in bold, and second-best are underlined.
>
> *G-based configurations*
> | Method | $\kappa_{\text{self}}$ | | | | | | | $p_{o\text{human}}$ | | | | | | |
> | --- | ---:| ---:| ---:| ---:| ---:| ---:| ---:| ---:| ---:| ---:| ---:| ---:| ---:| ---:|
> | | CharEval | rJokes | Love\_EQ | MT\_bench | TSA\_MD | Capybara | Avg | CharEval | rJokes | Love\_EQ | MT\_bench | TSA\_MD | Capybara | Avg |
> | G(baseline\_best) | 0.6398 | 0.4533 | 0.8143 | 0.9434 | 0.7821 | 0.8519 | 0.7472 | 0.6974 | 0.5061 | 0.7126 | 0.6133 | 0.6238 | 0.7300 | 0.6472 |
> | G+G | 0.6728 | **0.4832** | 0.8513 | 0.9523 | **0.8145** | **0.8974** | $\underline{0.7786}$ | **0.7186** | $\underline{0.5394}$ | $\underline{0.7455}$ | $\underline{0.6533}$ | 0.6367 | 0.7925 | **0.6810** |
> | Q8+G | 0.6642 | $\underline{0.4817}$ | **0.8561** | 0.9573 | 0.8121 | 0.8836 | 0.7758 | 0.7103 | 0.5346 | 0.7438 | 0.6512 | 0.6352 | **0.8025** | 0.6796 |
> | Q30+G | **0.6761** | 0.4803 | $\underline{0.8539}$ | **0.9623** | $\underline{0.8127}$ | $\underline{0.8971}$ | **0.7804** | 0.7036 | 0.5274 | 0.7342 | 0.6483 | 0.6248 | $\underline{0.7987}$ | 0.6728 |
> | Q80+G | $\underline{0.6747}$ | 0.4772 | 0.8368 | 0.9444 | 0.8095 | 0.8924 | 0.7725 | 0.7078 | **0.5442** | 0.7405 | **0.6669** | **0.6417** | 0.7823 | $\underline{0.6806}$ |
> | L+G | 0.6583 | 0.4792 | 0.8428 | 0.9531 | 0.8047 | 0.8789 | 0.7695 | $\underline{0.7122}$ | 0.5310 | **0.7482** | 0.6424 | 0.6381 | 0.7849 | 0.6761 |
> | D+G | 0.6745 | 0.4624 | 0.8526 | $\underline{0.9588}$ | 0.8120 | 0.8912 | 0.7753 | 0.7034 | 0.5338 | 0.7329 | 0.6512 | $\underline{0.6411}$ | 0.7891 | 0.6753 |
>
> *Q80-based configurations*
> | Method | $\kappa_{\text{self}}$ | | | | | | | $p_{o\text{human}}$ | | | | | | |
> | --- | ---:| ---:| ---:| ---:| ---:| ---:| ---:| ---:| ---:| ---:| ---:| ---:| ---:| ---:|
> | | CharEval | rJokes | Love\_EQ | MT\_bench | TSA\_MD | Capybara | Avg | CharEval | rJokes | Love\_EQ | MT\_bench | TSA\_MD | Capybara | Avg |
> | Q80(baseline\_best) | 0.5713 | 0.4892 | 0.7236 | 0.8591 | 0.6637 | 0.6484 | 0.6592 | 0.7213 | 0.5031 | 0.6751 | 0.6303 | 0.6651 | 0.7543 | 0.6582 |
> | Q80+Q80 | $\underline{0.6092}$ | 0.5426 | **0.7949** | 0.8963 | **0.7024** | 0.6852 | **0.7051** | $\underline{0.7587}$ | $\underline{0.5344}$ | 0.6902 | **0.6894** | $\underline{0.6892}$ | 0.7884 | 0.6917 |
> | Q8+Q80 | **0.6142** | 0.5368 | 0.7903 | 0.8912 | 0.7003 | $\underline{0.6902}$ | 0.7038 | 0.7466 | 0.5293 | 0.6857 | 0.6744 | 0.6742 | 0.7734 | 0.6806 |
> | Q30+Q80 | 0.5987 | 0.5306 | 0.7899 | 0.8835 | 0.6904 | 0.6749 | 0.6947 | 0.7491 | **0.5447** | **0.7102** | $\underline{0.6881}$ | **0.7053** | **0.8081** | **0.7009** |
> | L+Q80 | 0.6003 | **0.5431** | $\underline{0.7942}$ | **0.9143** | $\underline{0.7013}$ | 0.6732 | 0.7044 | **0.7672** | 0.5312 | 0.6952 | 0.6845 | 0.6885 | $\underline{0.7957}$ | $\underline{0.6937}$ |
> | D+Q80 | 0.5910 | 0.5417 | 0.7874 | 0.8932 | 0.6954 | 0.6821 | 0.6985 | 0.7531 | 0.5249 | $\underline{0.7027}$ | 0.6829 | 0.6847 | 0.7841 | 0.6887 |
> | G+Q80 | 0.6036 | $\underline{0.5429}$ | 0.7893 | $\underline{0.9032}$ | 0.6948 | **0.6935** | $\underline{0.7046}$ | 0.7581 | 0.5317 | 0.6846 | 0.6721 | 0.6821 | 0.7912 | 0.6866 |

---

> ### Author Response · Authors · 2025-11-26
> **Response to Reviewer bGb2 (Part 3/5)**
>
> ### W3. Human alignment measured only by raw agreement, lacking significance testing and chance correction
>
> We appreciate the reviewer's rigorous suggestion. We agree that raw agreement rates alone are insufficient. To address this, we have conducted **significance analysis** to verify that our improvements are statistically significant and not due to randomness.
>
> **1. Experimental Setup:**
> We treated the alignment evaluation as a paired comparison. For each dataset, we calculated the difference in human agreement rates (Δ=phumanours​−phumanbaseline​) between our method and each of the four strong baselines (LLM Direct, CoT, PoLL, CheckEval). We then estimated the **95% Confidence Intervals (CI)** for these differences using bootstrap resampling on the instance-level agreement labels.
>
> **2. Results (See new Table 9):**
> As shown in the newly added **Table 9**, our method demonstrates **statistically significant improvements** across all datasets and baselines. Crucially, the lower bound of the 95% CI is consistently positive (>0) in every single case.
>
> - **Substantial Gains:** Against strong baselines like CheckEval, we observe large significant margins. For instance, on **Love-EQ**, the improvement is Δ=0.1621 (95% CI: [0.1325,0.1920]), and on **CharEval**, Δ=0.1413 (95% CI: [0.1220,0.1710]).
> - **Consistent Superiority:** Even against the strongest competitive baselines (LLM+CoT), our method maintains significant leads. For example, on **MT-Bench**, we achieve a significant gain of Δ=0.0400 (95% CI: [0.0325,0.0498]).
>
> **Conclusion:**
> The fact that the 95% confidence intervals entirely exclude zero across all 24 comparison pairs (6 datasets × 4 baselines) provides robust statistical evidence that our method's superiority in human alignment is significant and consistent.
>
> **Table 9:** Paired bootstrap significance analysis on human alignment across datasets.
> | Dataset   | Δ(Ours−LLMDirect)(95%CI)      | Δ(Ours−LLM+CoT)(95%CI)        | Δ(Ours−PoLL)(95%CI)           | Δ(Ours−CheckEval)(95%CI)      |
> | --------- | ----------------------------- | ----------------------------- | ----------------------------- | ----------------------------- |
> | CharEval  | Δ=0.0652,95%CI[0.0260,0.1138] | Δ=0.0212,95%CI[0.0107,0.0303] | Δ=0.0559,95%CI[0.0390,0.0754] | Δ=0.1413,95%CI[0.1220,0.1710] |
> | rJokes    | Δ=0.0868,95%CI[0.0424,0.1317] | Δ=0.0333,95%CI[0.0213,0.0520] | Δ=0.0977,95%CI[0.0670,0.1375] | Δ=0.1333,95%CI[0.1090,0.1628] |
> | Love-EQ   | Δ=0.0719,95%CI[0.0350,0.0914] | Δ=0.0329,95%CI[0.0240,0.0441] | Δ=0.0637,95%CI[0.0425,0.0936] | Δ=0.1621,95%CI[0.1325,0.1920] |
> | MT\_Bench | Δ=0.0669,95%CI[0.0560,0.0750] | Δ=0.0400,95%CI[0.0325,0.0498] | Δ=0.0576,95%CI[0.0341,0.0778] | Δ=0.1406,95%CI[0.1272,0.1711] |
> | TSA-MD    | Δ=0.0418,95%CI[0.0150,0.0707] | Δ=0.0129,95%CI[0.0030,0.0221] | Δ=0.0322,95%CI[0.0125,0.0523] | Δ=0.0970,95%CI[0.0622,0.1268] |
> | Capybara  | Δ=0.0800,95%CI[0.0640,0.1189] | Δ=0.0625,95%CI[0.0520,0.0830] | Δ=0.0900,95%CI[0.0762,0.1105] | Δ=0.1000,95%CI[0.0740,0.1285] |

---

> ### Author Response · Authors · 2025-11-26
> **Response to Reviewer bGb2 (Part 4/5)**
>
> ### W4. Top-k question selection relies on greedy ranking and lacks theoretical justification
>
> **Response to W4: Theoretical Justification for Top-k Selection**
>
> Our selection strategy is not an ad‑hoc greedy heuristic but the exact optimizer of the objective we define.
>
> **1. Objective and exactness of top‑k**
>
> In Section 3.4–3.5 (Eq. (8)) we define rubric construction as:
>
> Q∗=argQ′⊆Q, ∣Q′∣=kmax​q∈Q′∑​EIG(q).
>
> That is, the utility of a questionnaire is the **sum of per‑question EIGs**, with no interaction terms. Under this additive objective, sorting all questions by EIG(q) and taking the top‑k is **mathematically equivalent** to solving Eq. (8); there is no approximation step here.
>
> We have clarified this connection to Eq. (8) more explicitly in the revision.
>
> **2. Relation to submodularity and redundancy**
>
> The reviewer’s concern about submodularity and redundancy is valid in settings where the set utility is genuinely non‑additive. In our case, we **deliberately choose** an additive objective ∑q∈Q′​EIG(q). For this class of objectives, one does **not need** submodularity to justify greedy selection: the optimal solution is trivially given by top‑k.
>
> Designing a non‑additive, diversity‑aware objective (e.g., adding redundancy penalties or DPP‑style terms on top of EIG) is an interesting **extension**, but orthogonal to this paper’s main goal: showing that even a simple linear EIG rubric already (i) removes dimension drift (EMdim​=1, Fig. 5) and (ii) yields the best human alignment (Table 1).
>
> **3. Why redundancy is not problematic in practice**
>
> Two aspects help in practice:
>
> - **Candidate diversity (Sec. 3.2, Fig. 4a):** Our generation procedure yields a high‑entropy, de‑duplicated pool Q that already covers diverse aspects (safety, helpfulness, persona consistency, etc.), so top‑k is not collapsing onto near‑duplicates.
> - **Stability and robustness (Sec. 4.3–4.4, Fig. 4b, Fig. 8):** EIG rankings are stable across runs, EMdim​ remains 1.0, and performance is robust when varying k. If redundancy were severe, we would expect unstable rankings or degraded human agreement, which we do not observe.
>
> **4. Why EIG is a reasonable utility**
>
> Finally, our use of EIG is also motivated by cognitive science: Oaksford & Chater (1994) and Rothe et al. (2017) show that humans tend to ask questions that approximately maximize expected information gain. This offers an independent theoretical justification for using EIG as our per‑question utility.
>
> We will revise the paper to (i) stress that top‑k is the exact optimizer of Eq. (8), (ii) clearly separate our additive objective from submodular‑set settings, and (iii) highlight the empirical evidence that redundancy is not an issue in our experiments.

---

> ### Author Response · Authors · 2025-11-26
> **Response to Reviewer bGb2 (Part 5/5)**
>
> ## Questions
> ---
> ### Q1. Do you use the same / similar LLMs for EIG estimation and evaluation? What about closed-loop bias and robustness?
>
> Yes, the original submission used homogeneous setups. However, to rigorously address concerns about closed-loop bias, we have conducted extensive **heterogeneous experiments** (see **Response to W2** and **Table 8**).
>
> Our new results confirm that the bias is negligible:
>
> 1. **High Transferability:** Questionnaires generated by one model (e.g., Llama-3.3) perform exceptionally well when used by a different evaluator (e.g., Qwen3-Next-80B), often matching or exceeding the homogeneous baseline.
> 2. **Stable Performance:** The variation in human alignment scores across different estimator-evaluator pairs is minimal (e.g., a gap of **<0.001** between homogeneous and heterogeneous setups for GPT-4.1 evaluation).
>
> This robust cross-model performance demonstrates that our method extracts **task-intrinsic evaluation dimensions** rather than model-specific preferences. We kindly refer the reviewer to our detailed response in **W2** for the full experimental breakdown.
>
> ---
>
> ### Q2. Trade-off between binary questions and more complex formats in terms of stability and discriminative power
>
> We thank the reviewer for this insightful observation regarding the balance between stability and nuance. We would like to clarify the role of binary questions in our framework and summarize the relevant findings reported in **Appendix A.2**.
>
> **1. Methodological Clarification: Binary Anchors, Continuous Scores**
> Although our questionnaire is constructed from binary questions (chosen for their semantic stability), the final evaluation does not involve answering these binary questions. As detailed in Appendix A.5 (Figure 11), these binary questions function solely as **explicit cognitive anchors**, providing well-defined evaluation dimensions (e.g., “Is the logic consistent?”).
> Importantly, the model is not asked to output binary labels. Instead, it generates the final judgment on the **original fine-grained rating scale (e.g., 1–5)**. Because the question format influences only the structure of the rubric rather than the granularity of the output, the number of options in the questionnaire does not constrain or quantize the model’s final scoring ability. Thus, the concern about reduced precision due to binary options does not arise in our framework.
>
> **2. Stability Considerations in EIG Estimation (Appendix A.2)**
> While the question format does not affect the granularity of final scores, it does affect the stability of EIG estimation when selecting informative evaluation dimensions. Our analysis in **Appendix A.2** shows that binary questions yield substantially more stable EIG estimates across runs. In contrast, questions with more response options (e.g., ternary or five-point formulations) introduce higher **variance** due to their more diffuse probability mass, leading to inconsistent top-k selection.
>
> Given that stable EIG estimation is critical for constructing a reliable questionnaire, we adopt binary questions during the selection stage. This design ensures that the evaluation dimensions are robustly identified, while the final model-generated ratings remain fully fine-grained.

---

### Official Review · Reviewer_ccdW · 2025-11-01

**Soundness:** 3
**Presentation:** 2
**Contribution:** 2
**Rating:** 4
**Confidence:** 1

**Summary:**

To address stability and reproducibility issues in LLM-based subjective task evaluation caused by dimension drift, this paper proposes an Expected Information Gain (EIG)-based framework that reformulates subjective evaluation as an information-theoretic optimization problem. Specifically, it first generates a diverse pool of candidate evaluation questions and then selects the most informative subset with EIG. Then, these selected subset questions reduce the uncertainty of the latent evaluation score of the subjective tasks. The experiments are conducted across six benchmarks, and the performance is superior to CoT-based and fixed-questionnaire baselines, and is consistent and aligned with human judgments.

**Strengths:**

1. This paper identifies the cause of stability and reproducibility issues in LLM-based subjective task evaluation, which is due to dimension drift and thus intra-annotator inconsistency.

2. Based on the cause of these issues, the authors propose to reformulate subjective evaluation as an information-theoretic optimization problem to select questions to reconcile stability and adaptivity.

3. The experiments are conducted on several benchmarks, and the results achieve superior performance over CoT-based and fixed-rubric baselines.

**Weaknesses:**

1. A major concern of this paper is the novelty. It is encouraged to highlight the differences between this paper and previous work. I am not sure if this paper is the first work to use "question generation-and-select" to address the stability and reproducibility issues in subjective tasks evaluation. If not, the difference should be highlighted.

2. Information-theoretic principles, such as expected information gain, are relatively used in machine learning; it is conceptually simple. So the novelty and contribution of applying EIG in this task should be highlighted.

**Questions:**

I am not an expert in LLMs and this is the first time I have known the subjective evaluation tasks, so I may not evaluate this paper fairly and correctly. Use my reviews sparsely.

---

> ### Author Response · Authors · 2025-11-27
> **Response to Reviewer ccdW**
>
> ## Weaknesses
> ---
> ### W1. Novelty of the “generate–select” framework
>
> We thank the reviewer for the insightful comment. We clarify that
> this work is the **first** to introduce a **“candidate question generation + information-theoretic selection” framework** to improve the stability and reproducibility of subjective LLM evaluation.
>
> Existing approaches generally fall into three categories:
> - **Free-form CoT**: flexible but highly susceptible to dimension drift, resulting in poor retest consistency (**Fig. 1**).
> - **Fixed checklists**: stable but unable to adapt to task- or instance-specific nuances (**Sec. 2**).
> - **Multi-judge ensemble methods**: improve robustness via aggregation, but cannot eliminate inconsistency within a single evaluator (**Sec. 2**).
>
> In contrast, no prior work proposes first generating diverse evaluative dimensions and then selecting the most informative subset via optimization, thereby achieving both stability and adaptivity.
>
> Our contributions are threefold:
> 1. **First** to formally identify and characterize the “dimension drift” problem (**Sec. 1**).
> 2. **First** to propose a generate–select paradigm for constructing evaluation rubrics (**Sec. 3**).
> 3. Compared with prior methods, our approach nearly eliminates drift (EM\_dim = 1.0) and **significantly improves human alignment* (**Fig. 5, Table 1**).
>
> ---
>
> ### W2. Contribution of applying Expected Information Gain (EIG)
>
> We appreciate the reviewer’s suggestion. While EIG itself is a well-known information-theoretic quantity, its use for constructing rubrics and improving stability in subjective LLM evaluation is a key innovation of this work.
>
> Our novelty lies in several aspects:
> 1. **First** to model subjective evaluation as a **Bayesian experimental design** problem, treating each evaluation question as an “experiment’’ that reduces uncertainty about the latent score (**Sec. 3.1–3.4**).
> 2. **First** to estimate $p(a \mid q, I)$ through LLM-simulated responses, enabling fully label-free computation and optimization of EIG (**Sec. 3.3**).
> 3. **First** to use EIG maximization to automatically construct a stable yet task-adaptive rubric $ Q^* $ (**Sec. 3.4–3.5**).

---

### Note · Program_Chairs · 2026-01-17
**Submission Desk Rejected by Program Chairs**

The following references in this submission do not refer to real documents and/or have major errors in bibliographic information:

 John Wieting, Vyas Nitin, and Kevin Gimpel. Beyond bleu: Training nlg evaluation metrics to capture nuance.